# AttTok: Marrying Attribute Tokens with Generative Pre-trained Vision-Language Models towards Medical Image Understanding

**Hualiang Wang, Xinyue Xu, Lehan Wang, Bin Pu**
Department of Electronic and Computer Engineering
The Hong Kong University of Science and Technology

**Xiaomeng Li** *
Department of Electronic and Computer Engineering
The Hong Kong University of Science and Technology
eexmli@ust.hk

## Abstract

Recent generative pre-trained vision–language (GPTv) models have achieved remarkable success in multi-modal understanding, inspiring their adaptation to medical imaging tasks such as disease diagnosis and visual question answering (VQA). However, current instruction-tuned GPTv models suffer from two key challenges: (1) medical attributes (*e.g.*, disease names, severity grades) are encoded as plain text tokens, collapsing semantically distinct concepts into nearly identical textual sequences; and (2) inadequate textual supervision weakens visual representation learning, leading to severe inter-attribute confusion and misaligned vision–language embeddings. To address these limitations, we introduce **att**ribute **tok**ens (AttTok), a set of pre-defined special tokens that uniquely encode clinical attributes (e.g., imaging modality, diagnosis, severity) within a structured token space. Complemented by attribute-centric embedding books, AttTok serves as anchor points for aligning both visual and textual modalities into a shared, discriminative representation space. Building on this foundation, we design two key components: an attribute-centric cross attention (ACC) adapter, which breaks the vision-to-text information-flow bottleneck and enriches the visual encoder with discriminative attribute knowledge, and an attribute-centric matching (ACM) loss, which enforces robust multi-modal alignment centered on the attribute tokens. Extensive experiments on five medical classification benchmarks and three VQA datasets demonstrate that AttTok substantially improves both discriminative accuracy and medical knowledge reasoning, establishing a new paradigm for medical GPTv models with clinically discriminative understanding. Codes.

## 1 Introduction

Recent advances in generative pre-trained vision–language (GPTv) models, such as GPT-4o (Hurst et al., 2024) and Qwen2.5-VL (Bai et al., 2025), have demonstrated remarkable progress across a wide range of universal visual understanding tasks (Peng et al., 2023; Liu et al., 2024a;b). These achievements have spurred growing interest in developing unified medical GPTv models, resulting in systems such as LLaVA-Med (Li et al., 2023), HealthGPT (Lin et al., 2025), and Lingshu (Xu et al., 2025). Despite their general capabilities, current medical GPTv models still fall short in disease diagnosis and classification tasks, particularly when required to deliver **precise decision-making** on extensive **clinical attributes** (*e.g.*, disease names and lesion types) across diverse medical imaging modalities, as evidenced by the limited zero-shot performance reported in Table 1.

Instruction tuning (Liu et al., 2023b) provides a feasible solution by representing medical attributes as **specialized textual terms** within instruction-tuning datasets. GPTv models learn to predict the

---

*Corresponding author

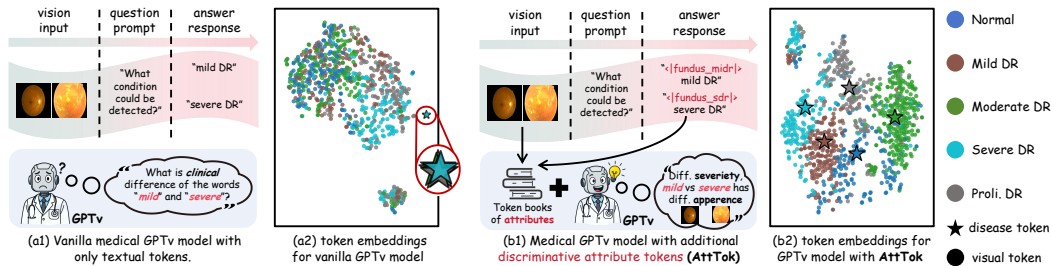

Figure 1: Comparison between **(a1)** vanilla medical GPTv with only textual response tokens and **(b1)** our proposed GPTv model with discriminative attribute tokens. **(a1)** The vanilla model produces plain textual answers (e.g., "mild DR", "severe DR"), which may lack explicit contextual or clinical meanings. **(b1)** By introducing attribute tokens and multi-modal token books, our model augments responses with discriminative features. **(a2)** and **(b2)** exhibit the embeddings of disease-level visual tokens (● with different colors) and disease tokens (★ with different colors) from two GPTv models instruction-tuned on DR grading dataset. The embeddings are projected into 2D space via t-SNE (Maaten & Hinton, 2008). The disease tokens are the textual disease names for vanilla GPTv models, and attribute tokens for GPTv models with AttTok.

textual sequence in a generative manner, *i.e.*, next-token prediction paradigm (Radford et al., 2018). Despite offering flexibility in representing diverse clinical concepts through word combinations, this paradigm is a double-edged sword, exposing two critical limitations:

**(1)** Encoding medical attributes as simple word combinations fails to reflect their underlying clinical semantics and inter-attribute distinctions. For instance, conditions such as *mild* diabetic retinopathy (DR) and *severe* diabetic retinopathy are represented by nearly identical textual sequences (Figure 1 (a1)) rather than as clinically meaningful categories with distinct pathological characteristics. As illustrated by the DR grading example in Figure 1 (a2), the token visualization reveals substantial confusion among textual tokens corresponding to different diseases, (marked via ★ with different colors and zoomed out here for clearer display) for the instruction-tuned medical GPTv model.

**(2)** More critically, owing to the causal information flow (Radford et al., 2018) in GPTv models (vision → text), inadequate textual encoding of medical attributes provides only weak supervision *back to* the visual modality. Consequently, the model fails to acquire discriminative visual representations required to capture clinically salient features. This shortcoming results in pronounced inter-attribute confusion within visual features and vision-text dis-alignment, as illustrated in Figure 1 (a2).

In essence, the generative paradigm provides flexibility but at the expense of **reducing the discriminative power of tokens**. As a result, it produces ambiguous representations of clinical attributes, entangled visual features, and weakened vision–language alignment. This raises a natural question:

> Can *discriminative* medical attributes be explicitly and precisely incorporated into the
> *generative* paradigm of GPTv models?

Motivated by this, we introduce a novel concept, **Attribute Tokens** (AttTok), a set of special tokens that uniquely convey the pre-defined clinical attributes. For instance, the token $<|fundus\_sdr|>$ specifies the imaging modality (fundus imaging) and the diagnostic finding (severe DR). Furthermore, attribute-centric embedding banks are constructed to provide representative features, **using attributes as anchors** to facilitate the alignment of multi-modal information.

On top of the attribute tokens and embedding books, **(1)** an attribute-centric cross attention (ACC) adapter is introduced to infuse clinical attribute knowledge from the embedding books into the visual encoder. ACC enhances visual saliency to perceive corresponding clinical attributes while eliminating the vision-to-text unidirectional bottleneck inherent in GPTv models. **(2)** an attribute-centric matching (ACM) loss is designed to enhance GPTv models by aligned and discriminative feature learning through intra-attribute consistency and inter-attribute separability.

To evaluate the effectiveness of our AttTok, we establish a comprehensive evaluation framework that systematically assesses both discriminative and generative capabilities of instruction-tuned GPTv models. The evaluation covers five publicly available datasets spanning diverse medical

imaging modalities, including dermatology (Derma), fundus photography (Fundus), optical coherence tomography (OCT), radiography (X-ray), and pathology (Path), addressing discriminative disease/lesion diagnosis and classification tasks. Furthermore, three commonly used medical VQA benchmarks, encompassing diverse medical knowledge, are employed to evaluate performance on medical knowledge understanding.

In summary, our contributions are:

**Conceptual:** We introduce attribute tokens and attribute-centric embedding books as explicit carriers of domain-specific attributes. These constructs provide GPTv models with structured access to discriminative medical semantics, thereby moving beyond surface-level word memorization.

**Technical:** We develop an attribute-centric cross attention adapter to highlight attribute-relevant visual embeddings, and an attribute-centric matching loss to align multi-modality information into a unified embedding space. These components jointly enable GPTv models to explicitly integrate discriminative medical knowledge into the generative paradigm, enhancing both semantic grounding and discriminative capacity.

## 2 RELATED WORK

### 2.1 GENERAL GPTv MODELS

The development of generative pre-trained vision–language (GPTv) models has drawn increasing attention in both computer vision and natural language processing communities. Broadly, prior efforts can be grouped into three threads.

**Large multimodal transformers.** Recent large-scale models such as GPT-4o (Hurst et al., 2024) and Qwen2.5-VL (Bai et al., 2025) integrate visual and textual streams within unified transformer architectures. These systems leverage massive web-scale corpora and joint optimization strategies to acquire strong perception, reasoning, and generation abilities across diverse modalities. A related line of work, such as Flamingo (Alayrac et al., 2022), Kosmos-1 (Huang et al., 2023), and PaLM-E (Driess et al., 2023), explores cross-modal fusion with frozen backbones, auxiliary adapters, or joint pretraining strategies, further demonstrating the scalability of multimodal LLMs.

**Instruction-tuned VL models.** Instruction tuning has proven to be an important technique for aligning multimodal models with user-centric tasks. Representative models include LLaVA (Liu et al., 2023a) and InstructBLIP (Dai et al., 2023), which adapt pretrained VL backbones with natural language instructions to support multi-turn dialogue, visual reasoning, and open-ended question answering. These approaches highlight the effectiveness of aligning general-purpose GPTv models with task-specific instructions while maintaining generalization capabilities.

**Task-specific extensions.** Beyond general reasoning, several models target structured multimodal tasks such as captioning, retrieval, or visual grounding. For example, PaLI (Chen et al., 2022) demonstrates unified performance on captioning and translation, whereas specialized models extend GPTv with external memory, retrieval-augmented features, or multi-granularity embeddings to improve interpretability or robustness. While successful in natural image understanding, these methods emphasize linguistic fluency and broad coverage rather than fine-grained attribute discrimination.

### 2.2 MEDICAL GPTv MODELS

The success of GPTv models in general domains has motivated their extension into medical applications, where multimodal reasoning is essential. This body of work can be divided into text-only and vision–language branches.

**Medical language models.** Early efforts such as BioGPT (Luo et al., 2022), PubMedGPT (Bolton et al., 2022), and PMC-LLaMA (Wu et al., 2024) focus on text-only corpora, capturing biomedical terminology and clinical reasoning through large-scale language modeling. More recently, Med-PaLM (Tu et al., 2024) and Med-PaLM 2 (Singhal et al., 2025) incorporate instruction tuning on curated QA datasets, yielding LLMs capable of expert-level responses in complex medical contexts.

**Multimodal medical GPTv models.** To further integrate visual signals, recent works such as Hua-Tuo (Wang et al., 2023), HealthGPT (Lin et al., 2025), and Lingshu (Xu et al., 2025) adapt multi-

modal instruction tuning pipelines to train GPTv variants on medical image–text pairs. These systems enable models to answer clinical visual questions, identify diseases, and provide textual explanations. In another direction, models such as ChatCAD (Tang et al., 2025) and Med-Flamingo (Moor et al., 2023) incorporate conversational interactions, retrieval mechanisms, or large-scale clinical datasets to improve alignment with radiology or pathology domains. These advances mark significant progress toward general-purpose medical GPTv assistants.

**Limitations.** Despite these efforts, most medical GPTv models inherit the training paradigm from general GPTv models, representing medical attributes (e.g., disease names, severity scales, anatomical structures) as plain text tokens. This introduces two fundamental limitations: (1) semantically distinct medical concepts are encoded as highly similar textual sequences, weakening discriminative capacity; and (2) the causal prediction objective offers weak supervision to the visual encoder, resulting in entangled features and ambiguous cross-modal alignment. Addressing these issues requires attribute-aware representations that explicitly capture clinical semantics while strengthening visual–textual alignment.

## 3 PRELIMINARY

The canonical generative pre-trained vision-language (GPTv) model is typically composed of three key components.

**(1) Vision encoder.** A vision encoder $\mathcal{E}_v$ (typically instantiated as a ViT Dosovitskiy et al. (2020)) maps an input image $\boldsymbol{I}$ into a sequence of visual token embeddings:

$$\boldsymbol{F}^v = \mathcal{E}_v(\boldsymbol{I}) \in \mathbb{R}^{N_v \times d}, \tag{1}$$

where $N_v$ denotes the number of visual tokens and $d$ is the embedding dimension.

**(2) Text tokenizer and embedding.** A tokenizer $\mathcal{T}$ converts input text into a sequence of discrete token indices, which are subsequently projected into the embedding space by a token embedding layer $\mathcal{E}_t$. The embedding layer maintains a set of token weights $\{\boldsymbol{e}_i\}_{i=1}^M$, with each $\boldsymbol{e}_i \in \mathbb{R}^{1 \times d}$ corresponding to the $i$-th entry in the vocabulary and $M$ indicating the total number of cadidate tokens. Given a word token $t$ from the textual questions $\boldsymbol{Q}$ or answer response $\boldsymbol{R}$, its index and embedding are computed as:

$$\begin{aligned} y &= \mathcal{T}(t), \\ \boldsymbol{f} &= \mathcal{E}_t(y) = \boldsymbol{e}_y, \end{aligned} \tag{2}$$

where $y \in \{1, \dots, M\}$ is the token index and $\boldsymbol{f} \in \mathbb{R}^{1 \times d}$ is the corresponding textual embedding, in terms of the word $t$ from $\boldsymbol{Q}$ and $\boldsymbol{R}$.

**(3) Decoder-only language model.** A decoder-only large language model $\mathcal{D}_l$ consumes a concatenated sequence of visual embeddings $\boldsymbol{F}^v$ (totally $N_v$ tokens), question embeddings $\boldsymbol{F}^q$ (totally $N_q$ answer token, calculated in Eqn 2), and previously generated response embeddings $\boldsymbol{F}^r_{<i}$ (tokens before the $i$-th token, calculated in Eqn 2). It predicts the probability distribution of the next answer token $y_i^r$ as:

$$p(y_i^r | \boldsymbol{I}, \boldsymbol{Q}, \boldsymbol{R}_{<i}) = \mathcal{D}_l(\boldsymbol{F}^v; \boldsymbol{F}^q; \boldsymbol{F}^r_{<i}), \quad i \in [1, N_r] \tag{3}$$

where $y_i^r$ is the ground-truth index of the $i$-th token in the response sequence with totally $N_r$ tokens.

**Training objective.** Training follows the next-token prediction paradigm, where the model is optimized with a cross-entropy loss over the answer sequence:

$$\mathcal{L}_{\text{NTP}}(\boldsymbol{R}) = -\frac{1}{N_r} \sum_{i=1}^{N_r} \log p(y_i^r | \boldsymbol{I}, \boldsymbol{Q}, \boldsymbol{R}_{<i}). \tag{4}$$

**Inference.** At test time, the sequence of the input image and question prompt is provided, and the model autoregressively generates the response tokens $y^r$ using the next-token prediction. This unified training-inference formulation ensures coherent answer generation.

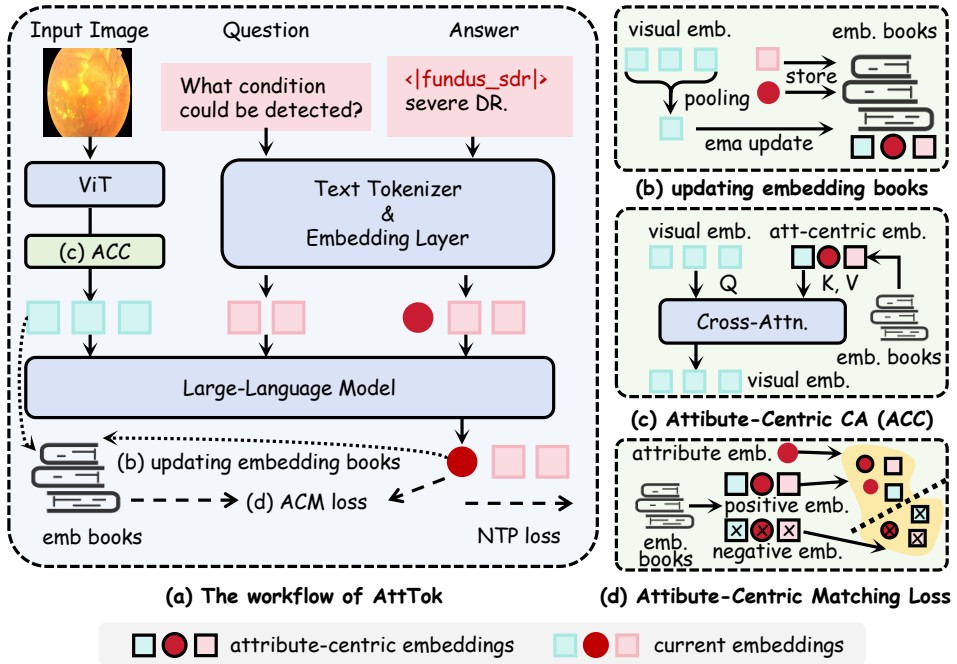

Figure 2: The architecture of AttTok. (a) The workflow integrates visual, textual, and attribute embeddings, processed through ViT, textual tokenizers, and embedding layers, and optimized with Next Token Prediction Loss and attribute-centric matching loss. (b) The updating strategy for the attribute-centric embedding books of pre-defined attributes. (c) The attribute-centric cross attention module enhances visual embeddings via discriminative embeddings from medical attributes. (d) The attribute-centric matching loss enforces alignment between positive attribute-centric embeddings while separating negative ones.

## 4 ATTRIBUTE TOKENS FOR GPTV MODELS

In this section, we introduce our framework for incorporating **clinical attribute tokens** into GPTv models. For each attribute token, we construct an embedding book that optimizes and stores discriminative visual, attribute and textual representations.

Building upon these token books, we further design two key modules: **(1) Attribute-Centric Cross Attention (ACC)** adapter, which enriches visual encoders with attribute-aware perception; and **(2) Attribute-Centric Matching (ACM)** loss, which enforces cross-modal alignment among visual, attribute, and textual tokens to better capture inter-attribute **clinical salience and distinctions**. The overall pipeline is illustrated in Figure 2.

### 4.1 INITIALIZING AND UPDATING ATTRIBUTE TOKENS AND TOKEN BOOKS

**Initializing Attributes.** Here, we formally define an **attribute token** as a new pre-defined token used to represent a specialized clinical concept. For example, $< |fundus\_sdr| >$ denotes that the input modality is fundus imaging and the associated clinical concept is severe diabetic retinopathy (DR). **In general, such concepts can correspond to a disease, a lesion, or an anatomic site.**

These attribute tokens are generated alongside the instruction-tuning datasets. **For classification-based datasets**, the attributes are naturally determined by the category labels. In contrast, **for VQA datasets**, the relevant attributes are pre-processed and extracted directly from the question–answer texts. Specifically, we employ open-source GPT models to summarize keywords from all question–answer pairs. Once the keywords are determined, we further use GPT to assign them to each question–answer pair, which serves as the ground-truth keyword annotation. These keywords are then organized into tokens following the format $< |modality\_concept| >$. For example, keyword

is "severe DR" and attribute token is $<|fundus\_sdr|>$ as aforementioned. **A detailed workflow is provided in Appendix A.2.**

**Initializing and Updating Attribute Token Books.** To incorporate attribute information into GPTv models, we expand the embedding layer to $\{e_i\}_{i=1}^{M} \cup \{a_i\}_{i=1}^{K}$, where $a_i$ is the token embedding of the $i$-th attribute token, with totally $K$ attribute tokens.

For each attribute token, we construct a multi-modal discriminative embedding book used each attribute as anchor. The embedding book for the $k$-th attribute is centered on its corresponding attribute token and contains: (1) the attribute token itself, $a_k$; (2) the textual token, $e_{\text{ind}(k)}$, corresponding to the attribute's keyword (where $\text{ind}(k)$ denotes the index of the textual tokens associated with attribute $k$); and (3) the visual prototype token, $\tilde{f}_k^v$, derived from the **average visual tokens** belonging to attribute $k$. Formally, $\mathcal{B}_k = \{a_k, e_{\text{ind}(k)}, \tilde{f}_k^v\}$.

To smoothly estimate the visual prototype for each attribute, we employ an exponential moving average (EMA) to iteratively update the prototype $\tilde{f}_k^v$. Given all visual embeddings associated with attribute $k$ in the current training batch, the visual prototype token is updated as

$$\tilde{f}_{k,new}^v = \mu \tilde{f}_{k,old}^v + (1-\mu)\frac{1}{N_v}\sum_{j \in N_v} f_j^v, \tag{5}$$

where $N_v$ denotes the number of visual embeddings belonging to attribute $k$ in the batch. $\mu$ is typically set as a value close to 1, Here we set it as 0.99. If no samples of attribute $k$ are present in current batch, the corresponding visual prototype remains unchanged. **Since both $a_k$ and $e_{\text{ind}(k)}$ are learnable parameters, they are jointly optimized during the training of the GPTv model.**

### 4.2 ATTRIBUTE-CENTRIC CROSS ATTENTION

In GPTv models, the causal information flow proceeds from vision to text, preventing the vision encoder from directly accessing textual information. However, pure textual tokens lack discriminative signals to effectively backpropagate supervision to the vision encoder. This limitation results in poor visual–text alignment.

To address this weakness, we propose an **Attribute-Centric Cross Attention** (ACC) adapter, which explicitly injects discriminative attribute knowledge into visual token representations. Essentially, the ACC module establishes a **skip route** that delivers attribute signals directly to the vision encoder.

Given the visual embeddings $F^v$ of the visual encoder, we enhance it via cross-attention over the attribute embedding books $\mathcal{B} = \{\mathcal{B}_k\}_{k=1}^{K}$ of all $K$ attribute. Formally, the total $3K$ tokens from $\mathcal{B}$ serve as the key and value embeddings and the visual embeddings $F^v$ is treated as query tokens, the adapted visual embeddings $\hat{F}^v$ are calculated as:

$$\text{Att}(F^v, \mathcal{B}) = \text{Softmax}\left(\frac{(F^v W_Q)(\mathcal{B} W_K)^\top}{\sqrt{d}}\right)(\mathcal{B} W_V), \tag{6}$$

$$\hat{F}^v = F^v + \gamma \text{Att}(F^v, \mathcal{B})W_O, \tag{7}$$

where $W_Q$, $W_K$, $W_V$ and $W_O$ are learnable projection matrices. $\gamma$ is a weighting factor that controls the strength of the ACC adapter. Here we set $\gamma = 0.1$. **Since direct access to ground-truth responses is unavailable in visual feature learning, all attributes are employed as both keys and values, enabling ACC to adaptively learn the correct activations.**

The enhanced representation $\hat{F}^v$ inherits discriminative guidance from both textual attributes and visual prototype tokens. This design enables the model to learn attribute-aware perception, strengthening the alignment between medical concepts and visual structures, and effectively **breaks the vision-to-text bottleneck** inherent in standard GPTv models.

### 4.3 ATTRIBUTE-CENTRIC MATCHING LOSS

While the cross-attention mechanism enriches visual tokens with attribute-aware information, it is still essential to explicitly enforce the alignment among visual representations, the corresponding

attribute tokens and their textual counterparts. To this end, we introduce an **Attribute-Centric Matching (ACM)** loss that leverages the attribute token books as anchors to drive discriminative and modality-aligned representation learning.

**Positive and Negative Sets.** Given an image, the GPTv model predicts the embedding vector $\boldsymbol{f}^a$ corresponding to the attribute $k$, The positive embedding book $\mathcal{B}_k$ provides three forms of positive supervision: (1) the attribute embedding, (2) the textual embedding of the attribute keywords, and (3) the visual prototype. Conversely, negatives are drawn from all other books $\mathcal{B}_j, j \neq k$.

**Similarity Measurement.** To quantify alignment, we employ a Linear layer, $\theta()$ ($\mathbb{R}^{1 \times d} \to \mathbb{R}^{1 \times d}$), to project embeddings into **a unified space** to measure the cosine similarity:

$$s(\boldsymbol{a}, \boldsymbol{b}) = \frac{\theta(\boldsymbol{a})^\top \theta(\boldsymbol{b})}{\|\theta(\boldsymbol{a})\|\|\theta(\boldsymbol{b})\|}. \tag{8}$$

This symmetric metric encourages tokens from the same attribute (regardless of modality) to lie close in the embedding space, while pushing tokens from different attributes apart.

**Multi-modality Matching Objective.** Building on this definition, we construct a matching loss that promotes alignment of the predicted attribute embedding $\boldsymbol{f}^a$ with all positives while contrasting it against negatives:

$$\mathcal{L}_{ACM}(\boldsymbol{f}^a) = -\log \frac{\sum\limits_{\boldsymbol{p} \in \mathcal{B}_k} \exp\left(\frac{s(\boldsymbol{f}^a, \mathbf{p})}{\tau}\right)}{\sum\limits_{\boldsymbol{p} \in \mathcal{B}_k} \exp\left(\frac{s(\boldsymbol{f}^a, \boldsymbol{p})}{\tau}\right) + \sum\limits_{\boldsymbol{n} \in \mathcal{B}_j; j \neq k} \exp\left(\frac{s(\boldsymbol{f}^a, \boldsymbol{n})}{\tau}\right)}, \tag{9}$$

where $\tau$ is a temperature parameter controlling concentration. Intuitively, this formulation performs soft classification of $\boldsymbol{f}^a$ with respect to the anchors in the attribute book. ACM loss integrates multi-modal (visual, textual, and attribute) supervision, thereby producing a richer and more discriminative signal for embedding alignment. Importantly, attribute books serve as consistent anchors across modalities, ensuring robust cross-modal coupling and disentanglement of distinct attributes.

Finally, the ACM loss is combined with the standard Next Token Prediction (NTP) loss to jointly optimize the model given an input sample $(\boldsymbol{I}, \boldsymbol{Q}, \boldsymbol{R})$. The total loss objective is $\mathcal{L}_{NTP} + \lambda \mathcal{L}_{ACM}$, where $\lambda$ is a hyperparameter balancing the contribution of the attribute-centric matching loss. The NTP loss ensures faithful generation of responses, while the ACM loss enforces discriminative attribute-aware representations, enabling the model to achieve both accurate medical understanding and coherent textual generation.

## 5 EXPERIMENT

### 5.1 EXPERIMENTAL DETAILS

**Baselines.** To comprehensively evaluate the performance of our AttTok on various medical benchmarks, we compare it against a diverse set of baseline models, encompassing both proprietary models and open-source GPTv models from both general and medical domains. For medical classification and diagnosis tasks, we further compare AttTok with CLIP-based models, the discriminative models. Specifically, our evaluation includes the following models. **(1) Medical GPTv** models include Med-R1 (Lai et al., 2025), MedVLM-R1 (Pan et al., 2025), HuatuoGPT-V (Wang et al., 2023), HealthGPT (Lin et al., 2025), LLaVA-Med (Li et al., 2023) and Lingshu (Xu et al., 2025). **(2) General-purpose GPTv** models consist of GPT-4.1 version Hurst et al. (2024) and Gemini-2.5-Flash (Comanici et al., 2025), which are the most representative proprietary models available with strong general visual understanding capabilities. Qwen2.5-VL-Instruct (Bai et al., 2025) and InternVL3 (Zhu et al., 2025), which are two state-of-the-art open-source models. **(3) CLIP-based models** include CLIP (Radford et al., 2021), SigLIP2 (Tschannen et al., 2025) and PubMedCLIP (Eslami et al., 2023). These discriminative vision-language models are used to give the comparision reference on medical classification tasks.

Table 1: Performance comparison of our methods with other GPTv models and fine-tuning methods on medical disease diagnosis and classification tasks across five modalities. The reported metrics are close-end and open-end accuracies. "−" **presents the unanswerable or extremely low performance on open-end zero-shot inference for precise disease diagnosis**.

| Models | Derma. | | Fundus | | OCT | | X-ray | | Path. | | Avg. | |
|---|---|---|---|---|---|---|---|---|---|---|---|---|
| | open | close | open | close | open | close | open | close | open | close | open | close |
| CLIP-based Discriminative Models | | | | | | | | | | | | |
| CLIP | 68.4 | | 62.6 | | 89.8 | | 93.5 | | 89.8 | | 80.8 | |
| SigLIP2 | 64.1 | | 65.3 | | 88.7 | | 92.9 | | 89.1 | | 80.0 | |
| PubMedCLIP | 65.1 | | 59.8 | | 89.6 | | 90.2 | | 88.2 | | 78.6 | |
| Zero-shot GPTv Models | | | | | | | | | | | | |
| HealthGPT-L | - | 13.9 | - | 28.4 | - | 29.1 | - | 65.9 | - | 34.7 | - | 34.4 |
| HuaTuoGPT-35B | - | 65.3 | - | 41.0 | - | 64.7 | - | 78.3 | - | 47.5 | - | 59.4 |
| Qwen2.5-VL-7B | - | 14.2 | - | 25.7 | - | 24.2 | - | 51.9 | - | 30.8 | - | 29.4 |
| Lingshu-7B | - | 41.9 | - | 33.2 | - | 25.5 | - | 85.0 | - | 56.8 | - | 48.5 |
| Instruction-Tuned GPTv Models | | | | | | | | | | | | |
| Qwen2.5-VL-7B | 65.8 | 71.2 | 55.0 | 61.5 | 59.1 | 73.0 | 76.9 | 87.3 | 72.4 | 81.7 | 65.8 | 74.9 |
| + Ours | 69.6 | 74.6 | 57.6 | 66.0 | 63.1 | 76.7 | 82.7 | 90.5 | 75.2 | 84.4 | 69.6 | 78.4 |
| Lingshu-7B | 66.3 | 72.8 | 56.3 | 63.7 | 60.8 | 74.7 | 78.9 | 89.1 | 73.5 | 84.3 | 67.2 | 76.9 |
| + Ours | 71.2 | 75.5 | 61.4 | 69.1 | 63.8 | 79.7 | 85.3 | 92.1 | 77.5 | 88.0 | 71.8 | 80.9 |

**Instruction tuning.** For all models, the default technique is LoRA which is a common-used adaptation for the parameter efficiency. Besides we also compare AttTok with different alignment techniques such as vision-text alignment CCL (Jiang et al., 2025) and visual-only alignment. For training data of AttTok, all attribute tokens are placed in front of the original text and serve as discriminative guidance.

**Benchmarks.** To validate the effectiveness of our proposed AttTok in enhancing medical image understanding tasks, we conduct experiments on two kinds of benchmarks.

Table 2: Performance comparison on medical VQA tasks.

| Model | Rad-VQA | SLAKE | PathVQA | Avg |
|---|---|---|---|---|
| Zero-shot GPTv Models | | | | |
| Med-R1-2B | 39.0 | 54.5 | 15.3 | 36.3 |
| MedVLM-R1-2B | 48.6 | 56.0 | 32.5 | 45.7 |
| LLaVA-Med-7B | 53.7 | 48.0 | 38.8 | 46.8 |
| HuatuoGPT-V-7B | 67.0 | 67.8 | 48.0 | 60.9 |
| Lingshu-7B | 67.9 | 83.1 | 61.9 | 71.0 |
| Instruction-Tuned GPTv Models | | | | |
| Qwen2.5VL-7B | 69.5 | 83.1 | 62.3 | 71.6 |
| + Ours | 70.1 | 84.0 | 63.5 | 72.5 |
| Lingshu-7B | 70.9 | 84.6 | 64.1 | 73.2 |
| + Ours | 71.4 | 85.8 | 64.7 | 74.0 |

**(1) Medical image diagnosis and classification tasks** are selected from five common medical imaging modalities, including ISIC-2018 (Milton, 2019) for seven-class skin lesion diagnosis. Optical coherence tomography (OCT) images (Kermany et al., 2018) for retinal diseases comprises of 4 diagnosis categories. NCT-CRC-HE-100K of histological images (Chen & Krishnan, 2022), comprises of 9 types of tissues. MessiDR (Lepetit-Aimon et al., 2024) is used to conduct DR grading task. X-Ray dataset (Yang et al., 2023) for pneumonia classifications is also employed.

**(2) Medical VQA tasks** include Rad-VQA (Lau et al., 2018), SLAKE (Liu et al., 2021), and PathVQA (He et al., 2020). These datasets cover CT, MRI, Chest-Xray, and Pathology modalities on questions involved location, modality and normal/abnormal judgment.

**Evaluation.** **(1) Diagnosis/Classification.** Datasets are organized as two formats including the open-ended questions (**without refereed category candidates in questions**), and close-ended questions (**with refereed category candidates in questions**).

**(2) VQA.** For **open-ended questions**, we first apply a strict, rule-based evaluation to check for exact matches (*i.e.*, accuracy) between the model prediction and the ground-truth answer. When the rule-based method does not find an exact match, an LLM-as-a-judge strategy is used to judge whether the prediction is semantically equivalent to the ground truth. For close-ended question, the rule-based accuracy is reported.

## 5.2 PERFORMANCE COMPARISON

**Medical Classification and Diagnosis.** Table 1 reports the results of medical classification across five modalities. We observe that: **(1) The Necessity of instruction tuning for precise medical imaging classification and diagnosis.** It is witnessed by the notably low zero-shot performance of GPTv models in this domain. **(2) Consistent and significant gains across general-purpose and medical-specific GPTv models.** This is evidenced by AttTok's consistent improvements across five modalities on both types of GPTv models, yielding at least a 2% increase in accuracy. **(3) Comparable to discriminative vision–language models.** Benefiting from the multi-modal discriminative capacity introduced by AttTok, GPTv models are able to achieve performance comparable to, and in skin lesion, DR grading and X-ray normal/abnormal status classification tasks surpassing, CLIP-based models, while still operating within a generative framework.

**Medical VQA.** The results on multi-modal VQA benchmarks are given in Table 2. The improvements from instruction tuning on Lingshu-7B (Xu et al., 2025) are relatively modest (compared to zero-shot version), this can be explained by the fact that Lingshu has already undergone extensive pretraining on large-scale medical data, including multiple rounds of VQA and report generation tasks, which leaves limited room for further gains. Nevertheless, even under this circumstance, our AttTok still achieves accuracy improvements via 0.5%, 1.2% and 0.6% accuracy improvement. This is particularly notable given that the attribute definitions and initializations (details in Appendix A.2) in VQA datasets are relatively coarse compared to the more explicit labels used in diagnostic and classification tasks.

## 5.3 ABLATION STUDY

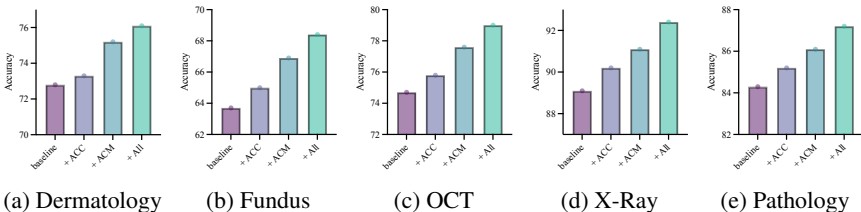

|  (a) Dermatology  |  (b) Fundus  |  (c) OCT  |  (d) X-Ray  |  (e) Pathology |

Figure 3: The effectiveness of ACC and ACM modules on five modality medical imagings.

**Effectiveness of ACC and ACM Modules.** Figure 3 reveals the effectiveness of the proposed components. (1) The attribute-centric cross attention adapter (ACC) breaks the bottleneck of existing GPT-v models, where information typically flows only from vision to text. By injecting attribute information into ViT, ACC enables richer cross-modal interaction and consistently boosts performance. (2) The attribute-centric matching loss (ACM) explicitly unifies clinical attributes in both visual and textual spaces, aligning them into a shared representation that captures domain-specific semantics. This design achieves substantially higher gains than using ACC alone. (3) When jointly applying ACC and ACM, the model achieves the strongest results, showing their complementary benefits and further validating the superiority of our attribute-centric alignment framework.

**Effectiveness on Different Token Alignment Strategies.** As shown in Table 4, both v.–t. alignment (directly aligning vision and text tokens, methods from (Jiang et al., 2025)) and v. alignment (clustering visual tokens) bring limited or unstable gains compared with the baseline. In particular, v. alignment even degrades performance on certain modalities (e.g., Fundus, OCT), indicating that naive visual token clustering loses critical fine-grained informa-

Table 3: Performance on different token alignment strategies.

| Method | Derma. | Fundus | OCT | X-Ray | Path. |
|---|---|---|---|---|---|
| Lingshu-7B ($r = 16$) | 72.8 | 63.7 | 74.7 | 89.1 | 84.3 |
| + v.-t. align. | 73.6 | 65.2 | 76.3 | 90.9 | 85.8 |
| + v. align. | 73.1 | 63.2 | 74.2 | 88.9 | 84.6 |
| + Ours | 76.1 | 68.4 | 79.0 | 92.4 | 87.2 |

tion. In contrast, our proposed strategy consistently outperforms all alternatives across five benchmarks, achieving the top accuracy on every modality (e.g., +3.3 on Dermatoscopy, +3.2 on Fundus, and +2.7 on OCT over the best baseline).

Additional experiments, including feature visualization analyses and extensions to imbalanced learning problems, are presented in Appendix A.3. A discussion of the limitations of our work and potential directions for future research can be found in Appendix A.5.

## 5.4 SCALING AND COMPUTATIONAL ANALYSIS ON MORE COMPLEX TASKS

Here, we further provide results on more challenging settings with more fine-grained or complex disease spaces: the Fitzpatrick17k (F17K) dataset with 113 fine-grained skin conditions, and the RFMiD dataset with 28 retinal conditions including comorbid combinations (diseases with less than 10 images are merged as other diseases). Note that for RFMiD, each image can be associated with multiple attribute tokens due to disease co-occurrence. We demonstrate the feasibility and necessity from two perspectives: (1) performance gains and (2) computational overhead.

To be specific, as the number of fine-grained diseases and comorbid combinations increases, the task becomes substantially more challenging, especially for GPT-v models, which are not inherently strong discriminative learners. Even the medically specialized Lingshu-7B model continues to struggle on these large, fine-grained label spaces after fine-tuning. In contrast, when augmented with our AttTok strategy, its performance improves substantially, from 27% to 41% accuracy on F17K and from 25% to 36% mAP on RFMiD, indicating that AttTok is particularly beneficial in precisely the regime of more complex disease diagnosis tasks. More importantly, with about 1-3% additional FLOPs introduced, the above performance improvements are obtained at nearly zero computational overhead.

Table 4: Performance on F17K and RFMiD.

| Models | Acc (F17K) | mAP (RFMiD) |
|---|---|---|
| CLIP-B16 | 0.40 | 0.45 |
| BioMedCLIP | 0.34 | 0.47 |
| PMC-CLIP | 0.45 | 0.49 |
| Lingshu-7B | 0.27 | 0.25 |
| AttTok | 0.41 | 0.36 |
| Improved. | 0.14 | 0.11 |
| New Flops | 2.5% | 1.4% |

For tasks with more than 1K diseases or concepts, assuming an input resolution of ($224 \times 224$), the vision encoder (VE) of Lingshu-7B/Qwen2.5-VL-7B requires approximately 340 GFLOPs. The LLM computation (which remains almost identical with and without AttTok) already exceeds 900 GFLOPs solely for encoding the visual tokens. The additional cost of the ACC module with (K) attributes is approximately ($(13 + 0.16K)$) GFLOPs. Thus, even when scaling to 1000 attributes, the computational overhead remains at merely 14% additional FLOPs.

## 5.5 SENSITIVITY ANALYSIS OF ATTRIBUTE KEYWORD QUALITY

**Robustness across different LLMs.** To assess the robustness of this process and the impact of potential keyword noise, we conduct an ablation in which three different LLMs (Qwen-3, InternLM, and GPT5) are independently used to derive the attribute book. The very small variance across LLMs (85.6%, 85.9% and 85.7%) indicates that the keyword extraction process is stable and not sensitive to the specific LLM used.

## 6 CONCLUSION

In this work, we identified two major limitations of instruction-tuned GPTv models for medical imaging: the inadequate encoding of clinical attributes as plain text tokens and the resulting inaccuracies in vision–language alignment. To overcome these challenges, we proposed **AttTok**, a unified framework that introduces structured attribute tokens, attribute-centric embeddings, and alignment mechanisms tailored for clinical concepts. Comprehensive evaluations across diverse benchmarks confirm its effectiveness, demonstrating consistent accuracy gains and improved medical reasoning.

## 7 ACKNOWLEDGMENTS

This work was partially supported by a research grant from the Research Grants Council (RGC) of Hong Kong (Project No. R6005-24); a grant from the Joint Research Scheme (JRS) jointly funded by the National Natural Science Foundation of China (NSFC) and the RGC of Hong Kong (Project No. N_HKUST654/24); additional funding from the RGC of Hong Kong (Project Nos. AoE/E-601/24-N and 16500825); and in part by a research grant from the National Natural Science Foundation of China (Grant No. 62306254).

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

## A APPENDIX

### A.1 EXPERIMENTAL DETAILS

For instruction tuning, we adopt LoRA (Hu et al., 2022) as the default adaptation technique with rank $r = 16$. The model is optimized using AdamW with a learning rate of $1 \times 10^{-4}$, weight decay of $1 \times 10^{-4}$, a cosine learning rate schedule, and a warmup phase over the first 10% of training iterations. We train with a batch size of 32 on 2 NVIDIA A800 GPUs for 3 epochs. The weight of the ACM loss is initially set to 1 and gradually decays to 0 during training. The $\tau$ in Eqn 9 is set as 1.

By default, both the vision tower and the LLM tower are frozen, and LoRA is only applied to linear layers. **To preserve the original encoding space of GPT-v**, we keep the embedding and classifier weights for all tokens except the newly introduced attribute tokens **unchanged** by zeroing their gradients. **This is implemented via PyTorch hook functions.**[1]

**The ablation study is conducted on Lingshu-7B by default, and the performance is evaluated on open-ended diagnosis and classification tasks.**

### A.2 WORKFLOWS FOR DATA PROCESSING AND ATTRIBUTE DEFINITION

**Dataset processing.** Given the complexity of GPTv training pipelines (especially built on the highly integrated Transformers (Wolf et al., 2020) and Llama-Factory (Zheng et al., 2024) libraries), it is impractical to perform online image augmentation as commonly used in conventional training for CLIP-based discriminative models (Radford et al., 2021). To address this, we adopt an **offline augmentation strategy** tailored for disease diagnosis/classification tasks. Specifically, we apply class-balanced sampling together with random cropping and horizontal flipping to generate augmented samples. Each augmented instance is stored as a new image, ensuring both diversity and balance in the training set while remaining compatible with the discriminative models.

**Attribute definition and assignment for VQA datasets.** Given a VQA dataset, we first input each sample's question, answer, and auxiliary information into Qwen-3 (Yang et al., 2025), and instruct it to reformulate the QA pair into a declarative sentence and extract key concepts as candidate keywords. After aggregating all candidates, we filter out non-discriminative terms (e.g., spatial positions, shapes, colors, sizes), redundant concepts, and low-frequency words. In other word, we only focus on words present-ing discriminative concepts, *i.e.*, modality, dis-

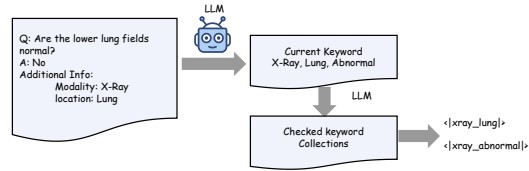

Figure 4: The workflow to generate the attribute words for samples from VQA datasets.

ease status (normal/abnormal, disease names), anatomical site. Given that we explicitly define the clinical concepts of interest, the scope of extracted keywords is constrained to a dozen categories. These keywords are restricted to **conclusive terms**, ensuring that the assigned keywords focus on diagnostically meaningful semantic.

The resulting refined keyword set is then used as a prompt, and Qwen-3 is further applied to assign appropriate keywords back to each sample. For PathVQA, since the dataset is exclusively composed of pathological images, the modality keyword attribute is omitted in this process.

### A.3 ADDITIONAL EXPERIMENTAL ANALYSIS

**The importance of attribute-centric alignment.** The Figure 5 presents the visualization of visual and attribute tokens across multiple datasets, demonstrating that the proposed *AttTok* substantially improves the discriminability of learned feature representations. We further examine two comple-mentary settings: (1) image generation using the AttTok-augmented Lingshu-7B model, and (2)

---

[1] https://docs.pytorch.org/docs/stable/generated/torch.Tensor.register_hook.html

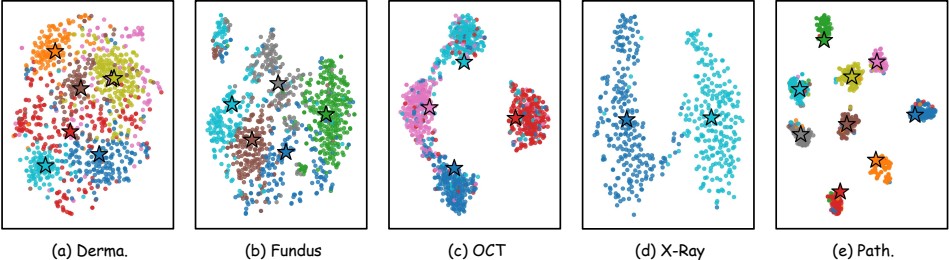

Figure 5: The feature visualization for five modalities, tuned via AttTok. Visual tokens (● with different colors) and attribute tokens (★ with different colors) are projected into 2D space via t-SNE (Maaten & Hinton, 2008). AttTok achieves clear attribute-centric multi-modality alignment.

feature retrieval with attribute tokens employed as classifier weights. As summarized in Table 5, attribute tokens provide a more accurate characterization of class-centered semantics. **Leveraging this property, AttTok can be utilized as a "teacher" module to distill discriminative capability into "student" GPTv models, thereby facilitating more robust multimodal alignment.**

| Cls Acc. | Derma. | Fundus | OCT | X-Ray | Path. |
|---|---|---|---|---|---|
| Retrieval via AttTok | 82.5 | 70.6 | 90.2 | 95.8 | 93.3 |
| GPTv with AttTok | 76.1 | 68.4 | 79.0 | 92.4 | 87.2 |

Table 5: Performance comparison between direct retrieval using attribute tokens as classifier weights and generative results from the fine-tuned GPTv model.

| Model | MEL (1113) | NV (6705) | BCC (514) | AKIEC (327) | BKL (1099) | DF (115) | VASC (142) |
|---|---|---|---|---|---|---|---|
| w/o AttTok | 35.8 | 82.2 | 69.2 | 25.4 | 25.4 | 11.8 | 48.7 |
| w/ AttTok | 53.3 | 85.4 | 70.4 | 35.4 | 35.4 | 31.4 | 52.2 |

Table 6: Lesion-wise classification performance (F1-score) on the ISIC-2018 (Milton, 2019) dataset. The numbers in parentheses denote the number of original training samples per category.

**Towards more discriminative diagnosis/classification tasks.** Here, we extend AttTok to tackle more complex and challenging discriminative diagnosis and classification tasks, imbalanced data problems which are commonly encountered in real-world clinical practice. As described in the dataset processing stage, we employ offline augmentation strategies to balance the training data. In this analysis, however, **we do not use pre-processing in order to expose the inherent imbalance** of the skin lesion classification dataset, ISIC-2018 (Milton, 2019). We report the class-wise F1 as Table 6. Except for the NV category, which dominates the dataset with an overwhelming number of training samples, most classes suffer from severe data scarcity. Under this highly imbalanced setting, the discriminative features introduced by *AttTok* substantially improve the F1 scores of all low-sample categories. **This observation highlights the potential of *AttTok* for handling the imbalanced data distributions that are commonly encountered in medical datasets.**

**Ablation on key hyper-parameters.** Here, we provide the ablation study on the scale of $\gamma$ and $\lambda$, testing on open-ended DR grading tasks. As shown in Table 7, larger $\gamma$ leads to over-fit on extra knowledge from attribute books, the empirical suitable values are in the range of [0.1, 0.5]. Similarly, larger $\lambda$ will hinder the learning of NTP and smaller one will shrink the impact of discriminative feature learning. A weight decay is needed.

| $\gamma$ | | | |
|---|---|---|---|
| $\gamma = 0.1$ | $\gamma = 0.5$ | $\gamma = 1.0$ | $\gamma = 2.0$ |
| 60.3 | 60.1 | 58.6 | 56.6 |

| $\lambda$ | | | |
|---|---|---|---|
| $\lambda = 0.1$ | $\lambda = 0.5$ | $\lambda = 1.0$ | $\lambda = 2.0$ |
| 58.2 | 57.5 | 60.3 | 59.8 |

Table 7: Ablation on $\gamma$ and $\lambda$.

## A.4 VISUALIZATION ANALYSIS ON ACC AND ACM

To illustrate the effectiveness of the proposed ACC module, a visualization analysis was conducted in Figure 6, which demonstrates the class activation maps for fea-

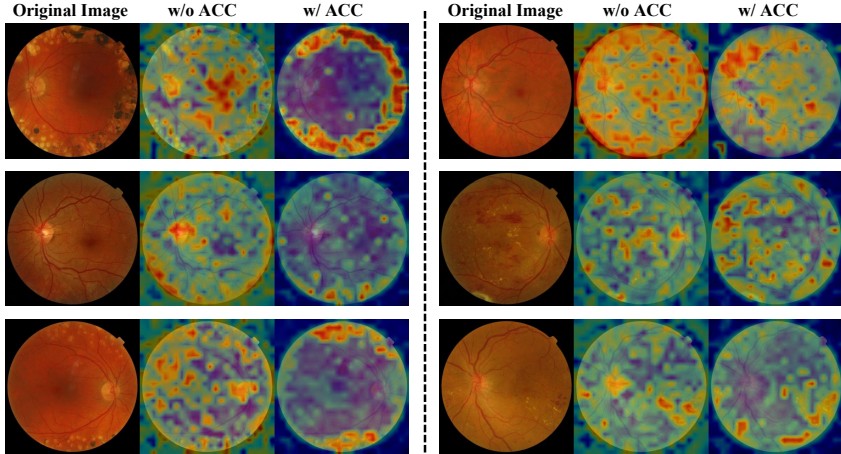

Figure 6: Visualization of ACC module's effectiveness. Left: ACC directs attention to correct visual cues (e.g., hemorrhage, exudates). Right: ACC mitigates erroneous background focus, despite not fully capturing foreground cues.

tures from visual encoder trained on DR grading tasks. This module utilizes prototype embeddings from attribute books to guide the visual encoder toward learning discriminative features. The left side of the visualization exhibits examples where the ACC module successfully directs the model's attention to appropriate visual cues, such as abnormal signals like hemorrhage and exudates. The right side demonstrates three cases where the ACC module effectively clears erroneous focus on the background, despite not yet achieving full concentration on correct foreground cues.

Furthermore, we analyze the feature correlations of the model trained without AttTok. When only textual features are used as guidance together with a purely causal information flow strategy, **the visual and textual signals for each pair of diseases become highly correlated and indistinguishable**. In contrast, training with AttTok explicitly guides the model through attribute tokens, which substantially enhances feature discriminability. At the classification level, this manifests as a clear reduction in confusion between different grading categories. To test class-wise metrics, we sample 100 testing images for each grading via data augmentation strategies.

## A.5 Discussion of Limitations and Future Explorations

**Text fluency, interpretability, and controllability.** As illustrated in Figure 1 and 2 and described in Section 4.1, all attribute tokens (multiple for VQA and multi-label classification tasks) are inserted in the front of the textual input sequence. In other words, the attribute tokens form a short, structured prefix that naturally transitions into the subsequent free-form text, without disrupting linguistic fluency or changing the conversational style. This design also makes the behavior of the model controllable: by learning attribute tokens firstly, we can explicitly steer the model's focus before it generates its response.

From an interpretability perspective, this setup leverages the causal information flow in GPT-v: tokens that appear earlier in the sequence causally influence the hidden states and attention distributions that determine all later tokens. Consequently, the attribute tokens act as explicit, discriminative guidance signals for the downstream text response. This is a core contribution of our work.

**Compositional attribute management.** Because one image can correspond to multiple questions, a single image may be associated with multiple attribute tokens. We treat these in a multi-label manner, allowing attributes to be combined orthogonally (e.g., organs, findings, and modalities together), which enables scaling to more attributes without conflict while preserving semantic completeness and flexible expression. This design ensures that our attribute space is both scalable and compositional, aligning with the multi-faceted nature of medical images.

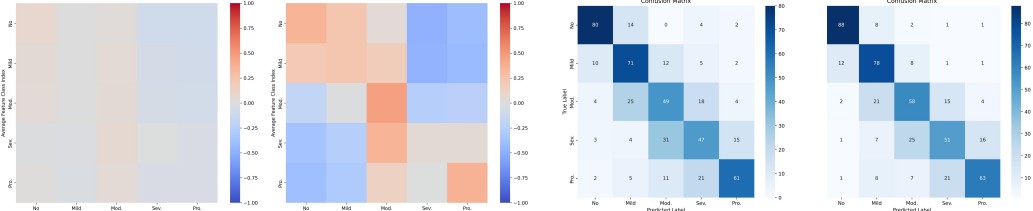

Figure 7: From left to right: The feature similarities for average visual features and textual features among five DR grading; the feature similarities for average visual features and attribute features among five DR grading; the confusion matrices for models trained w/o AttTok; the confusion matrices for models trained w/ AttTok.

**Limitations.** Although integrating discriminative attributes into GPTv models has been shown to enhance performance in diagnostic, classification, and VQA tasks, where short, conclusive answers are expected, the extension and validation of AttTok in long-text generation and fine-grained chain-of-thought reasoning tasks remain largely unexplored. Addressing these challenges will likely require not only the construction of AttTok but also more sophisticated attribute relations and even knowledge graph–based modeling, enabling AttTok to better handle longer and more complex textual scenarios.

**Future work.** In this study, we demonstrated the value of discriminative attributes for medical image diagnosis, thereby opening new research exploration on hybrid discriminative and generative representations on GPTv models.

Specifically, our findings point to the **potential of GPTv models in a broader range of discriminative tasks**, including (but not limited to) multi-label classification, few-shot learning, and long-tail problems. In parallel, they highlight promising opportunities to explore AttTok-based **unified frameworks in segmentation, detection, and other discriminative low-level tasks**.

Moreover, following the standard evolution route of GPTv models (Hurst et al., 2024; Bai et al., 2025; Xu et al., 2025), the stage after instruction tuning typically involves reinforcement learning–driven preference optimization and logical reasoning. Within this pathway, the AttTok framework presents promising opportunities for integration with online preference alignment strategies and the design of AttTok-informed reward models such as GRPO (Shao et al., 2024), paving the way for deeper explorations in task alignment and reasoning optimization. A feasible direction is to explore sentence-level modeling, extending AttTok from token-level discriminative representations to long-form representations. For example, reasoning-specific tokens could be introduced to guide a clearer chain-of-thought, enabling GPTv models to generate more structured and interpretable reasoning paths.

## B  LLM USAGE

Large Language Models (LLMs) were employed exclusively to assist with manuscript preparation. **Their role was limited to refining language and improving readability**. Specifically, the LLM contributed to tasks such as rephrasing sentences, correcting grammar, and smoothing the overall flow of the text.

The LLM played no part in formulating research questions, designing methodologies, conducting experiments, or analyzing results. **All scientific ideas, experimental designs, and analyses were conceived and executed entirely by the authors**. The authors accept full responsibility for the manuscript's content, including any text edited with LLM assistance. Care was taken to ensure that LLM-derived text complies with ethical standards and does not introduce plagiarism or scientific misconduct.

