# OpenReview forum: "AttTok: Marrying Attribute Tokens with Generative Pre-trained Vision-Language Models towards Medical Image Understanding"
_ICLR.cc/2026/Conference — ICLR 2026 Poster_

### Official Review · Reviewer_ZHpa · 2025-10-28

**Soundness:** 3
**Presentation:** 2
**Contribution:** 3
**Rating:** 6
**Confidence:** 4

**Summary:**

This study focuses on enhancing the fine-grained attribute discrimination capabilities of pre-trained GPTv models through "instruction tuning" (i.e., post-training). It addresses a limitation where existing GPTv models struggle with semantically distinct, fine-grained medical concepts due to their encoding as plain text tokens, leading to diminished discriminative power and misaligned vision-language embeddings. To overcome this, the authors propose **AttTok**, a framework that introduces specialized attribute tokens and associated attribute-centric embedding books for unique clinical attribute encoding. AttTok further integrates ACC adapter to infuse "discriminative attribute knowledge" into the visual encoder, and ACM loss to enforce robust "multi-modal alignment" centered on these attribute tokens. Extensive experiments across five medical classification benchmarks and three VQA datasets consistently demonstrate the effectiveness of AttTok. However, the analysis does not sufficiently clarify whether the observed performance gains genuinely stem from its attribute-centric design.

**Strengths:**

AttTok addresses the practical limitation that current GPTv models struggle with fine-grained medical attribute recognition, which is an important component of clinical reasoning.

- The proposed attribute-centric post-training strategy offers an efficient way to improve attribute sensitivity without modifying the base architecture.

- The combination of attribute tokens, embedding tables, an ACC adapter, and ACM loss is well motivated and leads to consistent performance gains across multiple medical benchmarks.

**Weaknesses:**

1. Ambiguity in Baseline Configuration: It is not clearly stated whether the "Instruction-Tuned GPTv Models" baselines (e.g., Lingshu-7B (r=16)) are trained on the exact same datasets as AttTok-enhanced models, but without AttTok’s attribute modeling components (attribute tokens, ACC adapter, ACM loss). Without explicit confirmation, one cannot fully exclude the possibility that performance gains originate from differences in the post-training setup rather than AttTok’s specific contributions.

2. Lack of Dedicated "Fine-grained Attribute Discrimination" Evaluation: Although AttTok aims to capture subtle inter-attribute distinctions, the paper does not provide a quantitative evaluation that directly assesses fine-grained discrimination capabilities. More detailed analyses such as confusion matrices or direct comparisons of highly similar clinical attributes (e.g., mild DR vs. severe DR) would help verify that AttTok truly improves sensitivity to fine-grained variations.

3. Limited Qualitative Evidence and Visual Insights: The paper lacks qualitative demonstrations that illustrate how AttTok influences attribute-aware perception within the visual encoder. Visualizations such as heatmap examples showing shifts in attention guided by attribute tokens could provide more interpretability. Case studies highlighting where baseline models fail but AttTok correctly identifies fine-grained findings would further reinforce the method’s clinical relevance.

**Questions:**

N/A

---

> ### Author Response · Authors · 2025-11-26
> **Response to Reviewer ZHpa**
>
> We thank the reviewers for recognizing the importance of the problem we address, as well as for acknowledging that our algorithm is well-motivated and both effective and efficient. We are also grateful for the reviewers’ valuable comments and suggestions.
> In the following, we clarify several points that may not have been sufficiently clear in the original manuscript and provide additional evidence to further support our claims.
>
> ---
>
> ## W1: Training configuration for baselines and AttTok.
>
> For clarity, we summarize our training setup and the differences from the baselines as follows.
>
> First, **all attribute tokens are placed in front of the original text** and serve as prepended discriminative guidance. Based on this setup:
>
> - Dataset. For all baseline methods, the training data consist of the original texts with the attribute prefixes removed.
> - Additional ACC module and ACM loss. These components are either simply enabled or disabled, without introducing any other changes.
> - Other training configurations. All remaining training hyperparameters are kept identical across methods, including the LoRA rank, the choice of layers to which LoRA is applied, the optimizer settings, and all other objective-related configurations.
>
> ---
>
> ## W2 and W3: More quantitative evaluation and qalitative evidence on capturing fine-grained discrimination capabilities.
>
> In the main paper, we provide Figure 5 and Table 5 to demonstrate the effectiveness of our method in enhancing attribute-centric discrimination and re-balancing imbalanced data distributions.
>
> Here, we additionally present class-wise confusion matrices on DR gradings to showcase the model’s ability to distinguish between easily-confused or visually similar diseases. Furthermore, we visualize the attention maps over visual features, attribute tokens, and disease names (i.e., visual–attribute–disease-name triplets), in order to provide a clear view of how the model’s visual activations and textual attention focus on the correct attributes.
>
> Since the newly proposed experiments from all four reviewers require additional time to run, we will upload a revised version of the main paper with the updated results as soon as they are available, and we kindly ask the reviewer to check the updated version at that time.
>
> ---
>
> We thank the reviewer again for their thoughtful feedback and believe that these clarifications and additional results address the raised concerns.

---

### Official Review · Reviewer_UJRS · 2025-10-30

**Soundness:** 2
**Presentation:** 3
**Contribution:** 2
**Rating:** 4
**Confidence:** 3

**Summary:**

The paper proposes AttTok, a unified framework to enhance medical GPTv models through learnable attribute tokens and embedding books that explicitly encode clinical concepts. It introduces an Attribute-Centric Cross Attention (ACC) adapter and Attribute-Centric Matching (ACM) loss to improve visual–text alignment, achieving consistent gains across multiple medical VQA benchmarks.

**Strengths:**

1) This manuscript clearly addresses the semantic collapse problem in medical GPTv models, motivation is clear.
2) The methodologically seems sound, with explicit formulation of attribute tokens, ACM loss, and prototype stabilization.
3) The end-to-end integration supporting both discriminative and generative paradigms.
4) Empirical evidence across diverse modalities and clear ablation validation supports the claims.

**Weaknesses:**

1) The approach builds on existing prototype-based and token-alignment ideas without substantial theoretical innovation, which limits its technical novelty
2) The method resembles prototype-guided learning. The paper should clarify how the attribute-token design and embedding book differ from prototype representations.
3) The paper lacks details discussion on the effort, criteria, and consistency in defining attributes, e.g., whether they are disease-specific or dataset-dependent. Furthermore, attribute tokens may encode explicit cues, inflating results.
4) Missing comparisons / discussions of the difference with existing medical multimodal models, such as BioMedCLIP and MedCLIP.
5) ACM loss lacks formal analysis or guarantees on improved alignment.
6) Evaluation is limited to classification and closed-form VQA, may perform additional tests on open-ended or noisy data.
7) Minor - next-token prediction paradigm (Radford et al.), lack of publication year.

**Questions:**

A few questions below, details refer to weakness.
1) How does AttTok differ conceptually and technically from prototype-based learning methods?
2) How are attribute tokens and embedding books defined—disease-specific or dataset-dependent—and what effort is required to construct them?
3) How sensitive is the model to attribute definition errors or GPT-generated keyword noise?
4) Can the authors provide ablation or scalability analysis on key hyperparameters (λ, γ) and large attribute sets?

---

> ### Author Response · Authors · 2025-11-26
> **Response to Reviewer UJRS**
>
> Thank you to the reviewers for their time, effort, and constructive comments, which have helped us significantly improve the paper.
>
> ## W1, W2 and Q1: The novelty statement.
>
> We highlight three **key differences** between AttTok and traditional **prototype learning**:
>
> 1. **Conceptual: from fixed prototypes to hybrid, more expressive tokens**
>
>    - Classic prototypes are **fixed class centers** used only for discrimination, each carrying **one static meaning**.
>    - AttTok’s **attribute tokens** are **hybrid discriminative–generative units**:
>      - they act as **discriminative anchors** in feature space,
>      - and are also **real text tokens**, which can appear and freely **combine** in natural language (e.g., modality, anatomy, disease subtype, comorbidity).
>    - This gives AttTok **far richer and more flexible semantics** than standard prototypes, which cannot be freely **composed** across prototypes.
>
> 2. **Technical: multimodal + joint training with NTP**
>
>    - We extend prototypes to a **multimodal** setting (visual features, textual features, and attribute anchors), instead of the usual **single-modality** prototypes.
>    - Attribute tokens are trained **jointly** with GPT-v under:
>      - attribute-related **discriminative losses**, and
>      - the **next-token prediction (NTP)** objective of the LLM.
>    - This tight integration with autoregressive language modeling is **not present** in conventional prototype learning.
>
> 3. **Role of prototype learning in AttTok**
>
>    - We **only borrow** a basic **update mechanism** (e.g., EMA-style updates) from prototype learning.
>    - In terms of **concept, loss design, and training strategy**, AttTok is fundamentally different: it aims to learn **controllable, composable attribute tokens** that guide GPT-v’s reasoning and generation, rather than just a set of static class centers.
>
> ---
>
> ## W3 and Q2: The generation criteria and pipeline of attribute keywords.
>
> As mentioned in **Section 4.1**, **Appendix A.2**, and **Figure 4**, we define and use attributes in a **strict and structured** way:
>
> 1. **Classification tasks: strictly defined attributes**
>
>    For classification tasks, attributes are **strictly defined** as:
>    > **unit of imaging modality + disease names**
>
>    That is, each attribute corresponds to a specific combination such as
>    *“chest X-ray + pneumonia”*, *“CT + intracranial hemorrhage”*, etc.
>    This avoids vague or overlapping labels and ensures that attributes are **well-grounded clinical units**.
>
> 2. **VQA tasks: structured info + conclusive keywords only**
>
>    For VQA, we use two sources of information:
>
>    - **Structured auxiliary information** provided in the dataset
>      (e.g., **anatomy**, **imaging modality**):
>      these are **directly added** as attribute words without LLM filtering.
>
>    - **Textual information in QA pairs**:
>      here, we use an LLM **only to select conclusive keywords**, only including
>        - disease / abnormality names,
>        - lesion or disease locations,
>        - organ names explicitly involved in the QA pair.
>
>      The LLM acts as a **selector**, not a generator: it filters from the given QA text and does **not** invent new attributes.
>
> 3. **Multi-label, compositional attribute management**
>
>    Because **one image can have multiple questions**, a single image may yield **multiple attribute tokens**.
>    We treat these in a **multi-label manner**, allowing attributes to:
>
>    - Be **combined orthogonally** (e.g., organs, findings, and modalities together),
>    - Scale to more attributes without conflict,
>    - Preserve **semantic completeness** and **flexible expression**.
>
>    This design ensures that our attribute space is both **scalable** and **compositional**, matching the multi-faceted nature of medical images.
>
>
> This strict attribute design criteria is crucial for AttTok: it yields **reliable, interpretable attribute tokens** that can be safely used as discriminative guidance for GPTv, without being polluted by noisy or non-conclusive text.

---

> > ### Author Response · Authors · 2025-11-26
> > **Response to Reviewer UJRS: Part 2**
> >
> > ## W4: Comparision to more medical multimodal models.
> >
> > We will add BioMedCLIP and MedCLIP as two additional CLIP-based discriminative baselines.
> >
> > In Table 1, where we currently compare CLIP, SigLIP, and PubMedCLIP, we will extend the table to also report BioMedCLIP and MedCLIP under the same evaluation protocol.
> > As suggested by Reviewer UEQz, we will update all results with mean ± standard deviation over multiple runs (instead of single-point estimates).
> > These new experiments and repeated runs require additional computation time.
> >
> > Once they are completed, we will update the main paper with: (1) the extended baseline comparison (including BioMedCLIP and MedCLIP), the mean ± std results for all main tables. (2) submit the revised version and kindly ask you to confirm the updated results and analysis.
> >
> > ---
> >
> > ## W5: The effectiveness of ACM loss.
> >
> > We provide the comprehensive feature analysis for models with and without ACM loss guidance, including T-SNE analysis (Figure 5). Besides, in Figure 3, using ACM loss solely leads to at least 2.2% accuracy improvement across five datasets. Figure 5 shows AttTok clearly learns attribute-centric discriminative features, an evidence for the performance gains.
> >
> > In the upcoming new revision of main paper, we also provide the attention map visualizations to highlight how ACM loss optimizes the relationships between multimodal cues (image regions and attribute tokens / text).
> >
> > Together, these qualitative and quantitative results provide strong evidence that ACM loss improves both discriminative power and interpretability of AttTok.
> >
> > ---
> >
> > ## W6: Evaluation on open-ended VQA tasks.
> >
> > We would like to clarify that Table 2 reports the overall accuracy for both close-ended and open-ended questions under our hybrid evaluation protocol (rule-based exact match + LLM-as-a-judge semantic matching).
> >
> >
> >
> > To complement this, we additionally provide **Recall for open-ended questions**, which is a classic rule-based metric purely based on string / word overlap between the prediction and the ground truth.
> >
> > | Model   | RAD-VQA | SLAKE | PathVQA |
> > |------|--------------|------------------|----------------|
> > | Qwen2.5-VL w/o AttTok   |    55.4  | 82.7     |    36.0    |
> > | Qwen2.5-VL w/ AttTok  | 59.5      | 84.1  |  37.4         |
> >
> >
> > ---
> >
> > ## Q3: Sensitivity analysis of attribute keyword quality in VQA tasks.
> >
> > To assess the robustness of this process and the impact of potential keyword noise, we conduct an ablation in which **three different LLMs** (Qwen-3, InternLM, and GPT5) are independently used to derive the attribute book.
> >
> > On the SLAKE VQA dataset, using attribute books from these different LLMs yields highly consistent performance over five independent runs:
> >
> >    - Qwen-3: 85.6 (±0.24)
> >    - InternLM: 85.9 (±0.27)
> >    - GPT5: 85.7 (±0.18)
> >
> > The very small variance across LLMs indicates that:
> > (a) the keyword extraction process is **stable** and not sensitive to the specific LLM used; and
> > (b) minor noise in the attribute keywords does **not propagate** in a way that meaningfully degrades the final VQA performance.
> >
> > ---

---

> > > ### Author Response · Authors · 2025-11-26
> > > **Response to Reviewer UJRS: Part 3**
> > >
> > > ## Q4: Scalability analysis on large attribute sets.
> > >
> > > We demonstrate the feasibility and necessity of AttTok in larger and more complex attribute spaces from two perspectives: (1) performance gains and (2) computational overhead.
> > >
> > > ### 1. Performance Gains
> > >
> > > To systematically evaluate scalability, we provide additional results in Fitzpatrick17k (F17K) dataset with 113 fine-grained skin conditions, and the RFMiD dataset with 28 retinal conditions, including comorbid combinations (diseases with fewer than 10 samples are merged into an "other" category).
> > >
> > > | Model               | Acc (F17K, 113) | mAP (RFMiD, 28) |
> > > |---------------------|-----------------|------------------|
> > > | CLIP-B16            | 0.40            | 0.45             |
> > > | BioMedCLIP          | 0.34            | 0.47             |
> > > | PMC-CLIP            | 0.45            | 0.49             |
> > > | Lingshu-7B          | 0.27            | 0.25             |
> > > | Lingshu-7B + AttTok | 0.41 (**Δ** 0.14) | 0.36 (**Δ** 0.11) |
> > >
> > > As the number of fine-grained diseases and comorbid combinations increases, the discriminative task becomes substantially more challenging. Even the medically specialized Lingshu-7B model struggles in such settings after standard fine-tuning. In contrast, when augmented with AttTok, its performance improves markedly: from 27% to 41% accuracy on F17K and from 25% to 36% mAP on RFMiD. These results confirm that **AttTok becomes increasingly essential as the attribute space grows in scale and complexity**.
> > >
> > > ### 2. Computational Evaluation
> > >
> > > We further analyze the computational overhead introduced by the Attribute Concept Coder (ACC) module and the Attribute Concept Modeling (ACM) loss.
> > >
> > > Given an input resolution of 224×224, the vision encoder (VE) in Lingshu-7B requires approximately **340 GFLOPs**. The LLM alone already consumes over **900 GFLOPs** for visual token encoding. The ACC module adds approximately (13 + 0.16*K*) GFLOPs, leading to the following scaling behavior:
> > >
> > > | K    | ~FLOPs (ACC) | ~FLOPs (VE+LLM) | Overhead ratio |
> > > |------|--------------|------------------|----------------|
> > > | 10   | 14.6G        | (340 + 900)G     | 1.2%           |
> > > | 100  | 29G          | (340 + 900)G     | 2.3%           |
> > > | 1000 | 173G         | (340 + 900)G     | 14.0%          |
> > >
> > > For the ACM loss, we introduce *K* additional attribute tokens, each associated with 3 exponential moving average (EMA) prototype vectors. Given that the base LLM (Lingshu-7B) has an embedding vocabulary of 152,064 tokens, the extra parameters introduced by ACM are negligible. For instance, with *K* = 1000, the parameter overhead is only **1.97%**.
> > >
> > > In summary, for **typical medical diagnostic settings**, which are usually subspecialty, focused and involve dozens to a few hundred categories, the computational overhead of AttTok is **minimal**. Even in **large-scale scenarios** with over **1,000 attributes**, the additional cost remains **well-controlled** (≈14% in FLOPs and <2% in parameters), and is modest relative to the base cost of the vision encoder and LLM.
> > >
> > > ---
> > >
> > > ## Q4: Ablation or scalability analysis on key hyperparameters.
> > >
> > > As described in Appendix A. 1 $\lambda$ is gradually decayed from 1 to 0. Here, we provide the ablation study on the scale of $\lambda$ and $\gamma$, testing on open-ended DR grading tasks.
> > >
> > >
> > > | $\gamma$ = 0.1  | $\gamma$ = 0.5 | $\gamma$ = 1.0 | $\gamma$ = 2.0 |
> > > |------|--------------|------------------|----------------|
> > > |  60.3 |    60.1 | 58. 6   |    56.6   |
> > >
> > >
> > > | $\lambda$ = 0.1  | $\lambda$ = 0.5 | $\lambda$ = 1.0 | $\lambda$ = 2.0 |
> > > |------|--------------|------------------|----------------|
> > > |  58.2 |    57.5 | 60.3   |    59.8  |
> > >
> > >
> > > In summary, larger $\gamma$ leads to over-fit on extra knowledge from attribute books, the empirical suitable values are in the range of [0.1, 0.5].
> > > Similarly, larger $\lambda$ will hinder the learning of NTP and smaller one will shrink the impact of discriminative feature learning. A weight decay is needed.
> > >
> > > Hyperparameters experiments with more datasets and larger value ranges will be added together in Appendix when all other experiments are finished.
> > >
> > > ---
> > >
> > > We hope these clarifications and additional results resolve all questions and concerns. We are happy to answer any remaining questions. Thank you for your detailed and thoughtful review.

---

### Official Review · Reviewer_AcFf · 2025-10-31

**Soundness:** 3
**Presentation:** 3
**Contribution:** 3
**Rating:** 6
**Confidence:** 4

**Summary:**

This paper addresses the failure of generative vision-language (GPTv) models in medical imaging, where encoding clinically distinct attributes (e.g., "mild" vs. "severe" disease) as similar plain text causes semantic ambiguity and poor vision-language alignment. The proposed solution, AttTok, introduces a set of pre-defined, special "attribute tokens" to uniquely represent these clinical concepts. This framework uses these tokens as anchors in two new components: an Attribute-centric Cross Attention (ACC) adapter to enrich the visual encoder with attribute knowledge, and an Attribute-centric Matching (ACM) loss to enforce robust multi-modal alignment via contrastive learning. Trained jointly with the standard next-token prediction loss, experiments across five medical classification and three VQA datasets demonstrate that AttTok substantially improves discriminative accuracy over baselines.

**Strengths:**

1. The paper clearly identifies and targets a core weakness of medical GPTv models: the poor discriminative ability of plain-text encodings for clinically distinct attributes.
2. The proposed solution is well-designed. The two main components, ACC and ACM, directly target the two identified weaknesses (information bottleneck and misalignment). The ACC adapter is an effective mechanism to inject top-down attribute knowledge into the visual encoder . The ACM loss is a well-founded contrastive objective to enforce multi-modal alignment
3. The evaluation is moderately thorough. The authors test on varied datasets covering diverse medical modalities.
4. The analysis in Table 5, showing that AttTok improves F1-scores for low-sample categories on an imbalanced dataset, is a strong point.

**Weaknesses:**

1. This is the most significant weakness. In Table 1, the average results for the Lingshu-7B + Ours model show a decrease in closed-ended accuracy (from 76.9% down to 71.6%) while improving open-ended accuracy (from 67.2% up to 80.6%). This directly contradicts the text, which claims "consistent and significant gains". This discrepancy is also seen in individual results (e.g., Lingshu-7B on Derma, where 'open' drops, and on Fundus, where 'close' drops). This confusion undermines the central claims and must be clarified.
2. The entire framework hinges on a set of "pre-defined special tokens". This raises critical questions about scalability. The method for defining attributes for VQA (using a GPT to extract keywords) seems ad-hoc and potentially brittle; the paper itself calls these attributes "relatively coarse". It is unclear how this approach would scale to thousands of fine-grained attributes or complex attribute combinations. The attribute books $\mathcal{B}$ would become very large, potentially making the ACC cross-attention and ACM matching computationally expensive.
3. As the authors acknowledge, the performance improvements on the three VQA benchmarks are modest (e.g., +0.8% average gain for Lingshu-7B). While the provided explanation (the baseline is already heavily pretrained) is reasonable, it limits the demonstrated impact of AttTok on tasks requiring more complex reasoning beyond classification.
4. The paper focuses on tasks with short, conclusive answers (classification and VQA). The authors concede in Appendix A.4 that the method's utility for long-text generation or chain-of-thought reasoning is "largely unexplored". It is unknown if adding the strong discriminative ACM loss harms the generative fluency, coherence, or nuance of the base LLM.

**Questions:**

1. Can you please address the apparent contradiction in Table 1? For the Lingshu-7B model, your method improves the average open-ended accuracy (67.2% $\rightarrow$ 80.6%) but seems to harm the closed-ended accuracy (76.9% $\rightarrow$ 71.6%). This contradicts the claim of "consistent improvements". Is this a typo in the table, or does AttTok introduce a trade-off between closed-ended and open-ended performance for this specific model?
2. The method relies on pre-defining $K$ attribute tokens. How do you see this approach scaling to a much larger and more complex attribute space (e.g., thousands of findings, locations, and modifiers)? Would the ACC module and ACM loss remain effective and computationally tractable as $K$ increases significantly?
3. The process for defining VQA attributes uses a GPT model to extract and assign keywords . How robust is this? Did you manually verify the quality of these extracted attributes? Given the "coarse" nature of these attributes, do you believe the modest gains on VQA are a limitation of the AttTok method itself or a result of noisy/imprecise attribute definitions?
4. Your method adds a strong discriminative objective (ACM loss) to the generative NTP loss. Did you perform any qualitative or quantitative analysis on whether this discriminative pressure harms the model's generative fluency, coherence, or its ability to generate nuanced, long-form explanations?

---

> ### Author Response · Authors · 2025-11-26
> **Response to Reviewer AcFf: Part 1**
>
> We sincerely thank the reviewer for the solid and constructive comments, which have significantly improved our work. Below, we first clarify several **misunderstandings**, and then address the reviewer’s **main concerns** with additional analyses and experiments.
>
>
> ## W1 and Q1: Mis-understanding on performance in Table 1.
>
> We wish to emphasize that there appears to be a **misunderstanding** regarding the results. Our method strictly dominates the baseline: 71.6% vs. 67.2% (open) and 80.6% vs. 76.9% (closed), indicating robust and significant superiority on both open-end and close-end classification tasks.
>
>
> ## W2 and Q2: Scalability analysis in larger and more complex attribute spaces.
>
> We demonstrate the feasibility and necessity of AttTok in larger and more complex attribute spaces from two perspectives: (1) performance gains and (2) computational overhead.
>
> ### 1. Performance Gains
>
> To systematically evaluate scalability, we provide additional results in Fitzpatrick17k (F17K) dataset with 113 fine-grained skin conditions, and the RFMiD dataset with 28 retinal conditions, including comorbid combinations (diseases with fewer than 10 samples are merged into an "other" category).
>
> | Model               | Acc (F17K, 113) | mAP (RFMiD, 28) |
> |---------------------|-----------------|------------------|
> | CLIP-B16            | 0.40            | 0.45             |
> | BioMedCLIP          | 0.34            | 0.47             |
> | PMC-CLIP            | 0.45            | 0.49             |
> | Lingshu-7B          | 0.27            | 0.25             |
> | Lingshu-7B + AttTok | 0.41 (**Δ** 0.14) | 0.36 (**Δ** 0.11) |
>
> As the number of fine-grained diseases and comorbid combinations increases, the discriminative task becomes substantially more challenging. Even the medically specialized Lingshu-7B model struggles in such settings after standard fine-tuning. In contrast, when augmented with AttTok, its performance improves markedly: from 27% to 41% accuracy on F17K and from 25% to 36% mAP on RFMiD. These results confirm that **AttTok becomes increasingly essential as the attribute space grows in scale and complexity**.
>
> ### 2. Computational Evaluation
>
> We further analyze the computational overhead introduced by the Attribute Concept Coder (ACC) module and the Attribute Concept Modeling (ACM) loss.
>
> Given an input resolution of 224×224, the vision encoder (VE) in Lingshu-7B requires approximately **340 GFLOPs**. The LLM alone already consumes over **900 GFLOPs** for visual token encoding. The ACC module adds approximately (13 + 0.16*K*) GFLOPs, leading to the following scaling behavior:
>
> | K    | ~FLOPs (ACC) | ~FLOPs (VE+LLM) | Overhead ratio |
> |------|--------------|------------------|----------------|
> | 10   | 14.6G        | (340 + 900)G     | 1.2%           |
> | 100  | 29G          | (340 + 900)G     | 2.3%           |
> | 1000 | 173G         | (340 + 900)G     | 14.0%          |
>
> For the ACM loss, we introduce *K* additional attribute tokens, each associated with 3 exponential moving average (EMA) prototype vectors. Given that the base LLM (Lingshu-7B) has an embedding vocabulary of 152,064 tokens, the extra parameters introduced by ACM are negligible. For instance, with *K* = 1000, the parameter overhead is only **1.97%**.
>
> In summary, for **typical medical diagnostic settings**, which are usually subspecialty, focused and involve dozens to a few hundred categories, the computational overhead of AttTok is **minimal**. Even in **large-scale scenarios** with over **1,000 attributes**, the additional cost remains **well-controlled** (≈14% in FLOPs and <2% in parameters), and is modest relative to the base cost of the vision encoder and LLM.
>
> ### 3. Conclusion
>
> Together, these analyses support two key claims:
>
> - **Scalability**: AttTok remains computationally feasible as the attribute space grows, with manageable increases in FLOPs and parameters.
> - **Necessity at Scale**: As the number and granularity of diagnostic attributes increase, AttTok becomes increasingly critical for enabling GPT-v models to learn discriminative representations capable of handling complex, fine-grained, and comorbid disease diagnoses.

---

> > ### Author Response · Authors · 2025-11-26
> > **Response to Reviewer AcFf: Part 2**
> >
> > ## W3 and W4: Utility for complex and long-text generation.
> >
> > Long-text generation, e.g., medical report generation (MRG), requires producing much longer texts that contain many **non-conclusive, sentence-level descriptive statements**. This makes the task fundamentally less dependent on discriminative signals. Our preliminary experiments confirm that AttTok’s **token-level discriminative modeling** brings only marginal gains for such long-form generation. Moreover, extracting reliable attribute keywords from long reports is substantially more difficult and noisy than from short QA pairs.
> > Concretely, on the **IU-Xray** dataset, we observe that adding AttTok to a strong MRG baseline yields statistically almost identical results (based on the mean over 5 independent runs):
> >
> > | Model                                | ROUGE-L | CIDEr |
> > |--------------------------------------|---------|-------|
> > | Qwen2.5-VL (sft, w/o AttTok)         | 29.32    | 80.6  |
> > | Qwen2.5-VL (sft, w/ AttTok)          | 29.13   | 80.9  |
> >
> >
> > The differences are within normal fluctuation and do not indicate a clear, consistent benefit. This empirically supports our design choice and claim:
> >
> > - For **short, conclusive tasks** (diagnosis, VQA), where **discriminative representations are crucial**, AttTok provides substantial performance gains and preserves coherence.
> > - For long-form report generation, which relies more on sentense descriptions, discriminative representation offers little advantage.
> >
> > Therefore, we focus our main contribution and analysis on tasks where AttTok’s **discriminative guidance** is most relevant and impactful.
> >
> > In the future, we plan to explore **sentence-level modeling** to extend AttTok from token-level discriminative representations to **long-form representations**. For example, a **reasoning-specific tokens** could be introduced to guide a clearer chain-of-thought, enabling GPTv models to generate more structured, interpretable reasoning paths.
> >
> >
> > ## W4 and Q4: Generative fluency, coherence, and nuance of the base LLM.
> >
> > As illustrated in Figures 1 and 2 and described in Section 4.1, all attribute tokens (multiple for VQA and multi-label classification tasks) are inserted in the front of the textual input sequence. In other words, the attribute tokens form a short, structured prefix that **naturally transitions into the subsequent free-form text**, without disrupting linguistic fluency or changing the conversational style. This design also makes the behavior of the model **controllable**: by learning attribute tokens firstly, we can explicitly steer the model’s focus before it generates its response.
> >
> >
> > Regarding **coherence**, in diagnosis and VQA tasks the model’s responses are typically **short and conclusive** (e.g., a specific disease name, anatomical site, or modality). In these settings, the insertion of attribute tokens at the beginning of the sequence does **not** harm coherence: the model produces consistent attribute judgement and textual conclusion, that is the attribute prefix naturally flows into the final answer.

---

### Official Review · Reviewer_UEQz · 2025-11-02

**Soundness:** 3
**Presentation:** 2
**Contribution:** 2
**Rating:** 4
**Confidence:** 4

**Summary:**

The paper proposes AttTok, a framework that introduces attribute tokens (e.g., <|fundus_sdr|>) and a multi-modal attribute dictionary (embedding book) containing textual keywords, learned embeddings, and EMA-updated visual prototypes.
Two main modules are designed:

* Attribute-Centered Cross-Attention (ACC): Allows visual features to cross-attend to attribute embeddings, breaking the traditional unidirectional “vision → text” flow in generative vision-language (GPT-v) models.

* Attribute-Centered Matching (ACM): A triplet-style objective aligning visual, textual, and attribute embeddings to improve intra-class consistency and inter-class separability.

**Strengths:**

* The attribute-token and dictionary act as explicit anchors; ACC and ACM target information flow and alignment respectively. The design is intuitive and easy to integrate into existing GPT-v pipelines (e.g., LoRA fine-tuning with frozen backbones).

* Demonstrates stable improvements across diverse medical modalities (fundus, dermoscopy, OCT, X-ray, pathology) and VQA tasks, showing strong generalizability and robustness even on advanced medical VL backbones.

**Weaknesses:**

* As the attribute space grows (rare diseases, fine-grained grades, comorbid combinations), the dictionary (~3K entries) and ACC computation may become computationally expensive. The paper lacks analysis of complexity, efficiency, or EMA prototype stability as K increases.

* For VQA, attribute keywords are extracted using Qwen-3. The paper does not assess label noise, propagation effects, or how mis-specified keywords may misguide alignment.

* Results mainly report accuracy without confidence intervals, AUROC, calibration, or significance testing. The use of LLM-based semantic scoring for VQA may introduce bias.

* It remains unclear how introducing attribute tokens affects text fluency, interpretability, or controllability, especially when encountering unseen or composite attributes.

**Questions:**

* Can new attributes (e.g., new diseases or grading levels) be added incrementally to the dictionary without retraining the entire model? How is representation drift prevented?

* Does AttTok handle combinations (e.g., “chest X-ray + mild infiltration + suspected heart failure”) via atomic tokens or compositional embeddings? How is combinatorial explosion managed?

* Since VQA accuracy is determined via LLM-based semantic matching, has the team validated with human or rule-based evaluation? How consistent are scores across evaluators?

* How does AttTok behave under domain shifts, image noise, or out-of-distribution (OOD) cases? Can attribute tokens support uncertainty estimation or refusal to answer?

---

> ### Author Response · Authors · 2025-11-25
> **Response to Reviewer UEQz: Part 1**
>
> We sincerely thank the reviewer for the thoughtful and constructive feedback. Your comments have helped us improve the clarity, practical value and impact of the manuscript. Below we address each questions in detail.
>
> ## W1 and Q2: Scalability analysis in larger and more complex attribute spaces.
>
> ### Performance gains
>
> We thank the reviewer for raising this important question regarding the scalability of our method to larger and more complex attribute spaces. In Tables 1 and 5 of the main paper, we analyze AttTok on datasets with imbalanced but relatively small disease spaces. Here, we further provide results on more challenging settings with **more fine-grained or complex disease spaces**: the Fitzpatrick17k (F17K) dataset with 113 fine-grained skin conditions, and the RFMiD dataset with 28 retinal conditions including comorbid combinations (diseases with <10 images are merged as other diseases). Note that for RFMiD, **each image can be associated with multiple attribute tokens** due to disease co-occurrence.
>
> | Model               | Acc (F17K, 113) | mAP (RFMiD, 28) |
> |---------------------|-----------------|------------------|
> | CLIP-B16            | 0.40            | 0.45             |
> | BioMedCLIP          | 0.34            | 0.47             |
> | PMC-CLIP            | 0.45            | 0.49             |
> | Lingshu-7B          | 0.27            | 0.25             |
> | Lingshu-7B + AttTok | 0.41 (**Δ** 0.14) | 0.36 (**Δ** 0.11) |
>
> As the number of fine-grained diseases and comorbid combinations increases, the task becomes substantially more challenging, especially for GPT-v models, which are **not inherently strong discriminative learners**. Even the medically specialized Lingshu-7B model continues to struggle on these large, fine-grained label spaces after fine-tuning. In contrast, when augmented with our AttTok strategy, its performance improves substantially, from 27% to 41% accuracy on F17K and from 25% to 36% mAP on RFMiD, indicating that AttTok is particularly beneficial in precisely the regime highlighted by the reviewer.
>
> ### Computational evaluation
>
> We now analyze the computational overhead introduced by the ACC module and ACM loss, and how they scale with the number of attributes \(K\).
>
> Assuming an input resolution of \(224 \times 224\), the vision encoder (VE) of Lingshu-7B/Qwen2.5-VL-7B requires approximately **340 GFLOPs**. The LLM computation (which remains almost identical with and without AttTok) already exceeds **900 GFLOPs solely for encoding the visual tokens**. The additional cost of the ACC module with \(K\) attributes is approximately **\((13 + 0.16K)\) GFLOPs**. Thus:
>
> | K    | ~FLOPs (ACC) | ~FLOPs (VE+LLM) | Overhead ratio |
> |------|--------------|------------------|----------------|
> | 10   | 14.6G        | (340 + 900)G     | 1.2%           |
> | 100  | 29G          | (340 + 900)G     | 2.3%           |
> | 1000 | 173G         | (340 + 900)G     | 14.0%          |
>
> For the ACM loss, we introduce \(K\) additional attribute tokens with 3 EMA prototype vectors each. Given that the embedding vocabulary size of the base LLM (Lingshu-7B) is 152,064, the extra linear weights introduced by ACM are negligible relative to the overall embedding parameters. For example, with \(K = 1000\), the parameter increase is only **1.97%**.
>
> In summary, for the **vast majority of disease-diagnosis scenarios**, which are typically subspecialty-focused and involve on the order of **dozens to a few hundreds of categories**, the additional computational cost of our method is **minimal**.  Even in **large-scale, general-purpose settings** involving more than **1,000** disease attributes, the overhead from ACC and ACM remains **well-controlled** (e.g., ≈14% in FLOPs and <2% in parameters), and is small compared to the base VE+LLM cost.
>
> ### Conclusion
>
> Overall, these analyses support two key conclusions:
>
> 1. **Scalability:** AttTok remains computationally feasible as the attribute/class space grows, both in terms of FLOPs and parameter count.
> 2. **Necessity at scale:** As the number and granularity of disease attributes increase, AttTok becomes increasingly important for enabling GPT-v models to acquire sufficiently discriminative representations for complex, fine-grained, and comorbid disease diagnosis.

---

> > ### Author Response · Authors · 2025-11-25
> > **Response to Reviewer UEQz: Part 2**
> >
> > ## W2: Sensitivity analysis of attribute keyword quality in VQA tasks
> >
> > We thank the reviewer for pointing out the potential issues of label noise, propagation effects, and mis-specified keywords in the VQA setting.
> >
> > For VQA tasks, we construct attribute keywords in a way that **minimizes label noise and semantic ambiguity**:
> >
> > 1. **Ground-truth–anchored input from short and concise Question-Answer pairs.**
> >    For each VQA sample, we feed the **exact question, the ground-truth answer, and structured auxiliary information** (e.g., anatomy, imaging modality, which are directly added as attribute words) into the LLM (Qwen-3 in the main experiments). The LLM is not asked to “understand” the image or to generate new content; instead, it operates on already-correct QA pairs and metadata.
> >
> > 2. **Rule-based selection, not free-form generation.**
> >    We explicitly design this as a **selection/filtering task** rather than an open-ended generation task. Qwen-3 is prompted to *select all and only the conclusive keywords* from the provided textual information, including disease/abnormal names, lesion/disease locations or organ names. Under this setup, the model does not invent new labels; it picks salient concepts already present in the question/answer/metadata. Consequently, each QA pair is associated with **multiple attribute tokens** that are all grounded in the annotated information, which substantially reduces the risk of mis-specified or hallucinated keywords.
> >
> > 3. **Robustness across different LLMs (noise and propagation).**
> >    To assess the robustness of this process and the impact of potential keyword noise, we conduct an ablation in which **three different LLMs** (Qwen-3, InternLM, and GPT5) are independently used to derive the attribute book:
> >    - The resulting conclusive attribute vocabularies are **almost identical** across the three LLMs in terms of the selected keywords, focusing on already provided structured keywords, disease names, locations and organs from QA texts.
> >    - On the SLAKE VQA dataset, using attribute books from these different LLMs yields highly consistent performance over five independent runs:
> >      - Qwen-3: 85.6 (±0.24)
> >      - InternLM: 85.9 (±0.27)
> >      - GPT5: 85.7 (±0.18)
> >
> >    The very small variance across LLMs indicates that:
> >    (a) the keyword extraction process is **stable** and not sensitive to the specific LLM used; and
> >    (b) any residual label noise in the attribute keywords does **not propagate** in a way that meaningfully degrades the final VQA performance.
> >
> > **Summary w.r.t. the reviewer’s concerns.**
> >
> > - **Label noise:** Mitigated by grounding keywords in the *gold* QA pairs and structured metadata, and by restricting the LLM to a selection task over provided information.
> > - **Focusing on conclusive words:** Only select the provided keywords from meta data, and targeted keywords, including disease name, lesions, locations and organs.
> > - **Mis-specified keywords misguiding alignment:** Our design (constrained selection + multiple conclusive tokens per QA pair) and the cross-LLM robustness study together suggest that AttTok is **insensitive to small perturbations** in the keyword set, and does not rely on any single fragile keyword to guide alignment.

---

> > > ### Author Response · Authors · 2025-11-25
> > > **Response to Reviewer UEQz: Part 3**
> > >
> > > ## W3 and Q3: LLM-based semantic scoring and more metrics for VQA tasks.
> > >
> > > ### Evaluation protocol: hybrid rule-based + LLM-as-a-judge
> > >
> > > We first apologize for the unclear and incomplete wording in lines 403–405 of the original manuscript.
> > >
> > > For VQA tasks, following the protocols in **Lingshu** and **MedEvalKit**, we adopt a **hybrid evaluation framework** that combines **rule-based matching** with an **LLM-as-a-Judge** step:
> > >
> > > 1. **Rule-based exact matching (primary).**
> > >    We first apply a strict, rule-based evaluation to check for **exact matches** between the model prediction and the ground-truth answer (including simple normalization such as lowercasing, punctuation handling, etc.). If an exact match is obtained, the answer is directly counted as correct, without involving any LLM. For **a GPTv with supervised instruction tuning**, this evaluation process is typically enough.
> > >
> > > 2. **LLM-based semantic adjudication (fallback).**
> > >    Only when the rule-based method does **not** find an exact match do we invoke an LLM to judge whether the prediction is **semantically equivalent** to the ground truth (e.g., “no abnormality” vs. “normal”). In this stage, the LLM is explicitly prompted as a **binary semantic comparator**. More importantly, answers in VQA tasks is typically **short** and **conclusive**, which is easy for LLM to judge.
> > >
> > >
> > > We adopt this strategy because traditional metrics (e.g., word-overlap Recall) can be misleading in the medical domain. For example, the pair **“mild diabetic retinopathy”** vs. **“moderate diabetic retinopathy”** achieves a high word-overlap Recall (0.66), despite being clinically distinct labels.
> > >
> > > ### Consistency and bias: human and multi-judge validation
> > >
> > > We acknowledge the reviewer’s concern that LLM-based scoring may introduce bias. To assess **consistency across evaluators**, we performed the following checks:
> > >
> > > - **Human vs. LLM agreement (spot-check).**
> > >   On a randomly sampled subset of VQA predictions, we compared the LLM-as-a-Judge decisions with **human decision**. We observed no wrong judgement  (we will include concrete comparisions in the supplementary), suggesting that the LLM’s semantic judgments on the short, answer-focused questions.
> > >
> > > ### Additional metrics for classification tasks.
> > >
> > > We also acknowledge the request for more comprehensive statistical reporting (confidence intervals and confusion matrices). In the current version, we mainly report point estimates of accuracy; during the rebuttal and revision, we will **expand** our reporting as follows:
> > >
> > > - For all experiments, we will report **mean ± standard deviation** over **5 independent runs**.
> > > - For disease diagnosis tasks, we will additionally provide **confusion matrices**, and report more visualization analysis from feature-level and decision-making perspectives.
> > > - For new MRG experiments, we already evaluate **ROUGE-L** and **CIDEr**, and will present these comprehensively in the final version.
> > >
> > > Considering that all updated experiments and repeated runs require extensive computation time, we will complete and report **mean ± standard deviation** over **5 independent runs** for the main results **within 5 days**.

---

> > > > ### Author Response · Authors · 2025-11-25
> > > > **Response to Reviewer UEQz: Part 4**
> > > >
> > > > ## W4: Text fluency, interpretability, and controllability.
> > > >
> > > > As illustrated in Figures 1 and 2 and described in Section 4.1, all attribute tokens (multiple for VQA and multi-label classification tasks) are inserted in the front of the textual input sequence. In other words, the attribute tokens form a short, structured prefix that **naturally transitions into the subsequent free-form text**, without disrupting linguistic fluency or changing the conversational style. This design also makes the behavior of the model **controllable**: by learning attribute tokens firstly, we can explicitly steer the model’s focus before it generates its response.
> > > >
> > > > From an interpretability perspective, this setup leverages the **causal information flow** in GPT-v: tokens that appear earlier in the sequence **causally influence** the hidden states and attention distributions that determine all later tokens. Consequently, the attribute tokens act as **explicit, discriminative guidance signals** for the downstream text response. This is a **core contribution** of our work.
> > > >
> > > >
> > > > ## Q1: The incremental learning paradigm for attribute tokens.
> > > >
> > > > This is an important question regarding incremental learning. We would like to clarify that the challenge of adding new knowledge without catastrophic forgetting is **a fundamental and inherent limitation of current GPT(v) models**, not a shortcoming specific to our method. This well-documented phenomenon, often termed "catastrophic forgetting,"[1][2] makes it exceptionally difficult for these models to incrementally update their knowledge base without performance degradation on previous tasks.
> > > >
> > > > To addressing this general limitation of GPT models is a critical research direction in its own right. Our AttTok framework, by providing a structured interface, actually creates a promising foundation for future work to develop specialized continual learning algorithms for such unified models.
> > > >
> > > > [1] Luo, Yun, et al. "An empirical study of catastrophic forgetting in large language models during continual fine-tuning." IEEE Transactions on Audio, Speech and Language Processing (2025).
> > > >
> > > > [2] Chen, Cheng, et al. "Coin: A benchmark of continual instruction tuning for multimodel large language models." Advances in Neural Information Processing Systems 37 (2024): 57817-57840.
> > > >
> > > > ## Q2: Attributes with combinations.
> > > >
> > > > To model attribute combinations, we formulate the problem in a **multi-label manner**, allowing multiple attributes to be present in a single image.
> > > > - In the context of disease diagnosis, these attributes correspond to co-existing conditions.
> > > > - For the VQA task, the diverse questions related to an image (e.g., about the disease, organ, or location) are treated as independent attribute labels, as illustrated in Figure 4.
> > > >
> > > > This formulation effectively enhances the combinatorial expressiveness of the attribute space, thereby avoiding a combinatorial explosion in the total number of attributes.
> > > >
> > > >
> > > > ## Q4: Generalizability on out-of-distribution data.
> > > >
> > > > This is a critical point that highlights a key contribution of AttTok: **it empowers GPT-series models with the novel capability to recognize and refuse out-of-distribution (OOD) visual questions**, "knowing when not to answer."
> > > >
> > > > To quantify this, we evaluated AttTok-equipped models using attribute-level confidence scores combined with standard OOD detection methods. We trained Lingshu model with AttTok on a Diabetic Retinopathy (DR) grading task and tested its OOD performance on the RFMiD dataset, treating non-DR images as OOD. This setup achieved an AUROC of 78.0%.
> > > >
> > > > In stark contrast, the instruction-tuned Lingshu model without AttTok lacks any OOD awareness. Being purely generative, it would invariably attempt to answer all queries, including OOD ones, leading to give DR gradings to all images. While a zero-shot GPT model can generate text beyond the DR domain, it does so indiscriminately and fails entirely at fine-grained medical diagnosis. Thus, AttTok is pivotal in enabling a reliable and safe diagnostic assistant.

---

> > > > > ### Comment · Reviewer_UEQz · 2025-11-28
> > > > >
> > > > > Thank you for your detailed response. I believe most of my questions have been addressed. I find this to be an interesting and valuable piece of work.

---

> > > > > > ### Author Response · Authors · 2025-11-28
> > > > > >
> > > > > > Thank you very much for your prompt and positive response. We are very glad to see that the most of your concerns have been addressed. All of the above content, as well as the new experiment results, will be incorporated into the main paper within one day. **If you have any further points you would like to discuss, we would be very happy to continue the conversation**.
> > > > > >
> > > > > > In addition, although the system currently disallows score edits, we would like to kindly ask whether you would be willing to consider increasing your score for this paper if editing is enabled at a later stage.
> > > > > >
> > > > > > Best regards,
> > > > > > AttTok Authors

---

### Author Response · Authors · 2025-11-29
**General Response Summary**

We sincerely thank the reviewers for their constructive and valuable comments.


## Strength summary

Generally, all reviewers agree that the problem addressed in this work is **“meaningful”, “practical”, and a “core weakness”**. The proposed method is **"well-motivated" and "well-designed"**. **“Promising” and “thorough”** experimental results across diverse modalities, together with clear ablation studies, support our claims.

Besides,
- Reviewer `UEQz` thinks our method is easy to integrate into existing GPT-v pipelines, showing **strong generalizability and robustness** even on advanced medical VL backbones.
- Reviewer `AcFf` thinks the method has the effective mechanism and well-founded objective. The evaluation is moderately thorough. **The effectiveness on imbalanced datasets is a strong point**.
- Reviewer `UJRS` thinks the methodology is sound and appreciates the end-to-end integration **supporting both discriminative and generative paradigms**.
- Reviewer `ZHpa` thinks it offers an efficient way to improve attribute sensitivity **without modifying the base architecture**.


## Concerns and discussion summary

We have provided detailed, point-by-point responses to all raised concerns. Following the discussion phase, Reviewer `UEQz` acknowledged **that most of their questions have been successfully addressed** and affirmed that this paper is **an interesting and valuable piece of work**.
Due to the abruptly concluded rebuttal phase, we regret that the other reviewers were unable to participate in the discussion.

The following summarizes the specific point-by-point questions and our corresponding responses.


**Common** concerns:

- c1. Scalability analysis in larger and more complex attribute spaces. (Reviewer `UEQz, AcFf, UJRS`)
    - AttTok provides more significant performance improvements on fine-grained tasks with 100+ diseases and comorbid combinations.
    - AttTok remains computationally feasible as the attribute space grows, with manageable increases in FLOPs and parameters.

- c2. Sensitivity analysis of attribute keyword quality in VQA tasks. (Reviewer `UEQz, UJRS`)
    - New experimental results show that the designed keyword extraction process is stable and not sensitive to the specific LLM used.

- c3. Generative fluency, coherence, interpretability, and controllability. (Reviewer `UEQz, AcFf`)
    - AttTok acts as explicit, discriminative guidance signals, inserted at the beginning of the text, to enhance interpretability without disrupting the original fluency.

**Specific** concerns:

- s1. Generalizability on out-of-distribution data. (Reviewer `UEQz`)

    - We empirically demonstrate that AttTok empowers GPT-series models with the **novel capability** to recognize and refuse out-of-distribution (OOD) visual questions, effectively "knowing when not to answer."

- s2. More rule-based metrics and LLM-based scoring analysis. (Reviewer `UEQz`)
    - We add robust analysis and new evaluation metrics.

- s3. Application to incremental learning. (Reviewer `UEQz`)
    - We clarify that this concern falls outside the scope of the current work and is considered a subject for future research.

- s4. Evaluation on long-text generation tasks. (Reviewer `AcFf`)

    - We provide new experimental results and discussions on extending AttTok to long-text generation tasks in future work, such as moving from **attribute tokens** to **reasoning-specific tokens**.

- s5. Clarification of the differences between AttTok and traditional prototype learning. (Reviewer `UJRS`)
    - We discuss the conceptual distinctions, technical details, and relationships between AttTok and prototype-based methods.

- s6. Comparison with more medical CLIP models. (Reviewer `UJRS`)

    - We include new experimental results with additional medical CLIP baselines.

- s7. Ablation on key hyperparameters. (Reviewer `UJRS`)
    - We add new ablation results on critical hyperparameters.

Besides, one misunderstanding has been clarified and some detailed descriptions have been added into the main paper:

- Misunderstanding of the content of Table 1. (Reviewer `AcFf`)
- Generation criteria and pipeline for attribute keywords. (Reviewer `UJRS`)
- Training configurations for baselines and AttTok. (Reviewer `ZHpa`)
- More qualitative evidence. (Reviewer `ZHpa`)

---

### Meta-Review · Area_Chair_XvRR · 2026-01-02

**Summary:**

This paper proposes AttTok, a simple yet effective framework that introduces attribute tokens to mitigate semantic collapse in generative vision–language models for medical image understanding. Reviewers consistently found the problem important and the approach well motivated, with strong empirical results across multiple medical imaging benchmarks.

The main concerns raised during review focused on scalability to large or compositional attribute spaces, robustness of LLM-extracted attributes, clarity of novelty relative to prototype-based representations, and limited gains in long-form generative outputs. The rebuttal addressed these points convincingly through additional experiments and analyses, including scalability and FLOPs evaluations, cross-LLM robustness studies, clearer positioning against related methods, and expanded diagnostic analyses (e.g., confusion matrices and attention visualizations). While improvements in long-text generation remain modest and theoretical grounding is limited, these limitations are clearly acknowledged and do not undermine the core contribution.

Overall, the work is technically sound, clearly presented, and offers a practically useful design that can be adopted in medical VLM systems. The contribution is incremental but solid, and the experimental evidence supports the claims.

**Reviewer Concerns:**

The rebuttal convincingly addressed most concerns, including scalability (with both FLOPs analysis and large-attribute experiments), robustness of attribute extraction (cross-LLM ablations), clarification of novelty vs. prototypes, and baseline fairness. Additional analyses on OOD detection, confusion matrices, and attention visualizations strengthened the claims. Remaining limitations—most notably limited benefits for long-text generation and lack of theory—are acknowledged and non-blocking.

**Reviewer Scores:**

Reviewer UEQz: Likely increase from 4 → 6 (explicitly stated concerns were resolved).

Reviewer AcFf: Likely unchanged (6) after clarification of Table 1 and scalability.

Reviewer UJRS: Likely raised given improved novelty and robustness explanations.

Reviewer ZHpa: Likely unchanged (6), with added evidence reinforcing attribution of gains.

---

### Decision · Program_Chairs · 2026-01-26

Accept (Poster)